# LEARNING SUPER-FEATURES FOR IMAGE RETRIEVAL

**Philippe Weinzaepfel, Thomas Lucas, Diane Larlus, and Yannis Kalantidis**
NAVER LABS Europe, Grenoble, France

## ABSTRACT

Methods that combine local and global features have recently shown excellent performance on multiple challenging deep image retrieval benchmarks, but their use of local features raises at least two issues. First, these local features simply boil down to the localized map activations of a neural network, and hence can be extremely redundant. Second, they are typically trained with a global loss that only acts on top of an aggregation of local features; by contrast, testing is based on local feature matching, which creates a discrepancy between training and testing. In this paper, we propose a novel architecture for deep image retrieval, based solely on mid-level features that we call *Super-features*. These Super-features are constructed by an iterative attention module and constitute an *ordered set* in which each element focuses on a localized *and* discriminant image pattern. For training, they require only image labels. A contrastive loss operates directly at the level of Super-features and focuses on those that match across images. A second complementary loss encourages diversity. Experiments on common landmark retrieval benchmarks validate that Super-features substantially outperform state-of-the-art methods when using the same number of features, and only require a significantly smaller memory footprint to match their performance.

Code and models are available at: https://github.com/naver/FIRe.

## 1 INTRODUCTION

Image retrieval is a task that models exemplar-based recognition, *i.e.* a class-agnostic, fine-grained understanding task which requires to retrieve all images matching a query image over an (often very large) image collection. It requires learning features that are discriminative enough for a highly detailed visual understanding but also robust enough to extreme viewpoint/pose or illumination changes. A popular image retrieval task is landmark retrieval, whose goal is to single out pictures of the exact same landmark out of millions of images, possibly containing a different landmark from the exact same fine-grained class (*e.g.* 'gothic-era churches with twin bell towers').

While early approaches relied on handcrafted local descriptors, recent methods use image-level (global) or local Convolutional Neural Networks (CNN) features, see Csurka & Humenberger (2018) for a review. The current state of the art performs matching or re-ranking using CNN-based local features (Noh et al., 2017; Cao et al., 2020; Tolias et al., 2020) and only learns with global (*i.e.* image-level) annotations and losses. This is done by aggregating all local features into a global representation on which the loss is applied, creating a discrepancy between training and inference.

Attention maps from modules like the ones proposed by Vaswani et al. (2017) are able to capture intermediate scene-level information, which makes them fundamentally similar to mid-level features (Xiao et al., 2015; Chen et al., 2019). Unlike individual neurons in CNNs which are highly localized, attention maps may span the full input tensor and focus on more global or semantic patterns. Yet, the applicability of mid-level features for instance-level recognition and image retrieval is currently underwhelming; we argue that this is due to the following reasons: generic attention maps are not localized and may fire on multiple unrelated locations; at the same time, object-centric attentions such as the one proposed by Locatello et al. (2020) produce too few attentional features and there is no mechanism to supervise them individually. In both cases, methods apply supervision at the global level, and the produced attentional features are simply not discriminative enough.

In this paper, we present a novel image representation and training framework based solely on attentional features we call *Super-features*. We introduce an iterative Local feature Integration Trans-

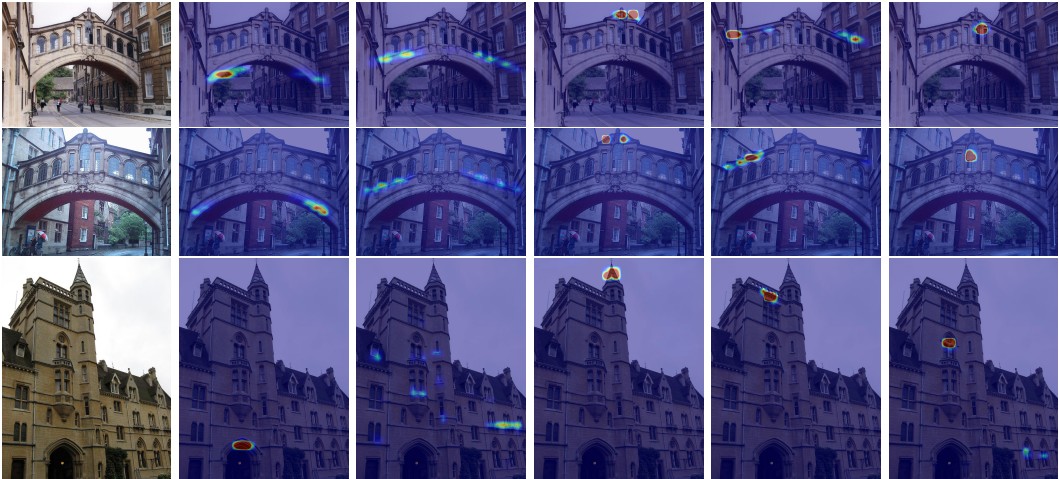

Figure 1: **Super-features attention maps** produced by our iterative attention module (LIT) for three images (left), with the first two that match, for five Super-features. They tend to consistently fire on some semantic patterns, *e.g.* circular shapes, windows, building tops (second to fourth columns).

former (LIT), which tailors existing attention modules to the task of image retrieval. Compared to the slot attention (Locatello et al., 2020) for example, it is able to output an ordered and much larger set of features, as it is based on learned templates, and has a simplified recurrence mechanism. For learning, we devise a loss that is applied directly to Super-features, yet it only requires image-level annotations. It pairs a contrastive loss on a set of matching Super-features across matching images, with a decorrelation loss on the attention maps of each image, to encourage Super-feature diversity. In the end, our network extracts for each image a fixed-size set of Super-features that are semantically ordered, *i.e.*, each firing on different types of patterns; see Figure 1 for some examples.

At test time, we follow the protocol of the best performing recent retrieval methods and use ASMK (Tolias et al., 2013), except that we aggregate and match Super-features instead of local features. Our experiments show that the proposed method significantly outperforms the state of the art on common benchmarks like $\mathcal{R}$Oxford and $\mathcal{R}$Paris (Radenović et al., 2018a), while requiring less memory. We further show that performance gains persist in the larger scale, *i.e.* after adding 1M distractor images. Exhaustive ablations suggest that Super-features are less redundant and more discriminative than local features.

**Contributions.** Our contribution is threefold: (a) an image representation based on *Super-features* and an iterative module to extract them; (b) a framework to learn such representations, based on a loss applied directly on Super-features yet only requiring image-level labels; (c) extensive evaluations that show *significant* performance gains over the state of the art for landmark image retrieval. We call our method *Feature Integration-based Retrieval* or FIRe for short.

## 2 BACKGROUND: LEARNING LOCAL FEATURES WITH A GLOBAL LOSS

Let function $f : \mathcal{I} \rightarrow \mathbb{R}^{W \times H \times D}$ denote a convolutional neural network (CNN) backbone that encodes an input image $\boldsymbol{x} \in \mathcal{I}$ into a $(W \times H \times D)$-sized tensor of $D$-dimensional local activations over a $(W \times H)$ spatial grid. After flattening the spatial dimensions, the output of $f$ can also be seen as set of $L = W \cdot H$ feature vectors denoted by $\mathcal{U} = \{\boldsymbol{u}_l \in \mathbb{R}^D : l \in 1 .. L\}$; note that the size of this set varies with the resolution of the input image. These local features are then typically whitened and their dimension reduced, a process that we represent by function $o(\cdot)$ in this paper. Global representations, *i.e.* image-level feature vectors, are commonly produced by averaging all local features, *e.g.* via global average or max pooling (Babenko & Lempitsky, 2015; Tolias et al., 2016; Gordo et al., 2016).

**A global contrastive loss for training.** Tolias et al. (2020) argue that optimizing global representations is a good surrogate for learning local features to be used together with efficient match kernels for image retrieval. When building their global representation $g(\mathcal{U})$, they weight the contribution of

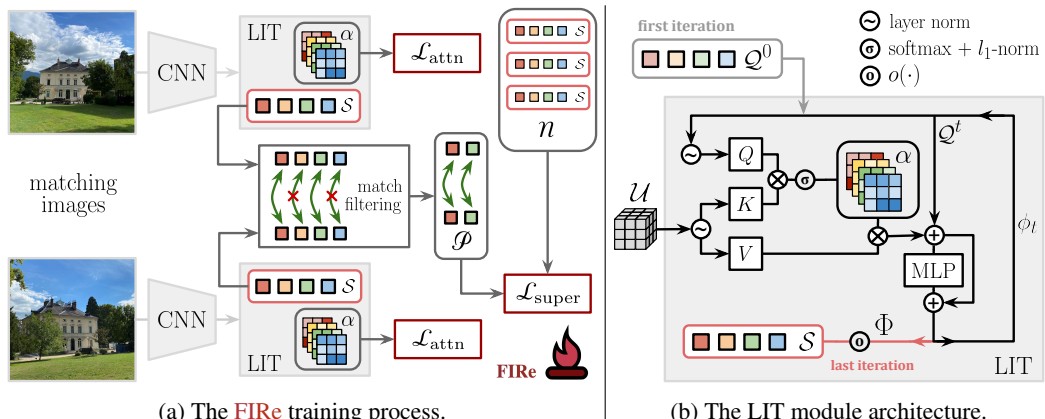

(a) The FIRe training process.

(b) The LIT module architecture.

Figure 2: **An overview of FIRe.** Given a pair of matching images encoded by a CNN encoder, the iterative attention module LIT (Section 3.1) outputs an ordered set of Super-features. A filtering process keeps only reliable Super-feature pairs across matching images (Section 3.2§1), which are fed into a Super-feature-level contrastive loss (Section 3.2§2), while a decorrelation loss reduces the spatial redundancy of the Super-features attention maps for each image (Section 3.2§3).

each local feature to the aggregated vector using its $l_2$ norm:

$$ g(\mathcal{U}) = \frac{\hat{g}(\mathcal{U})}{\|\hat{g}(\mathcal{U})\|_2}, \qquad \hat{g}(\mathcal{U}) = \sum_{l=1}^{L} \|\boldsymbol{u}_l\|_2 \cdot o(\boldsymbol{u}_l), \tag{1} $$

where $\|\cdot\|_2$ denotes the $l_2$ norm. Given a database where each image pair is annotated as matching with each other or not, they minimize a contrastive loss over tuples of global representations. Intuitively, this loss encourages the global representations of matching images to be similar and those of non-matching images to be dissimilar. Let tuple $(\mathcal{U}, \mathcal{U}^+, \mathcal{V}_1^-, \ldots, \mathcal{V}_n^-)$ represent the sets of local features of images $(\boldsymbol{x}, \boldsymbol{x}^+, \boldsymbol{y}_1^-, \ldots, \boldsymbol{y}_n^-)$, where $\boldsymbol{x}$ and $\boldsymbol{x}^+$ are matching images (*i.e.* a positive pair) and none of the images $\boldsymbol{y}_1^-, \ldots, \boldsymbol{y}_n^-$ is matching with image $\boldsymbol{x}$ (*i.e.* they are negatives). Let $[\cdot]^+$ denote the positive part and $\mu$ a margin hyper-parameter. They define a contrastive loss over global representations as:

$$ \mathcal{L}_{global} = \left\| g(\mathcal{U}) - g(\mathcal{U}^+) \right\|_2^2 + \sum_{j=1}^{n} \left[ \mu - \left\| g(\mathcal{U}) - g(\mathcal{V}_j^-) \right\|_2^2 \right]^+. \tag{2} $$

In *HOW*, Tolias et al. (2020) employ the global contrastive loss of Eq.(2) to learn a model whose local features are then used with match kernels such as ASMK (Tolias et al., 2013) for image retrieval. ASMK is a matching process defined over selective matching kernels of local features; it is a much stricter and more precise matching function than comparing global representations, and is crucial for achieving good performance. By learning solely using a loss defined over global representations and directly using ASMK over local features $\mathcal{U}$, HOW achieves excellent image retrieval performance.

## 3 LEARNING WITH SUPER-FEATURES

Methods using a global loss for training but local features for matching have a number of disadvantages. First, using local activations as local features leads to high redundancy, as they exhaustively cover highly overlapping patches of the input image. Second, using ASMK on local features from a model trained with a global loss introduces a mismatch between training and testing: the local features used for ASMK are only trained *implicitly*, but are expected to individually match in the matching kernel. To obtain less redundant feature sets, we propose to learn *Super-features* using a loss function that operates directly on those features, and to also use the latter during retrieval; the train/testing discrepancy of the pipeline presented in Section 2 is thus eliminated.

In this section, we first introduce the Local feature Integration Transformer (*LIT*), an iterative attention module which produces an ordered set of Super-features (Section 3.1). We then present a framework for effectively learning such features (Section 3.2) that consists of two losses: A contrastive loss that matches individual Super-features across positive image pairs, and a decorrelation

loss on the Super-feature attention maps that encourages them to be diverse. An overview of the pipeline is depicted in Figure 2. We refer to our approach as *Feature Integration-based Retrieval* or FIRe for short, an homage to the feature integration theory of Treisman & Gelade (1980).

## 3.1 Local feature Integration Transformer (LIT)

Inspired by the recent success of attention mechanisms for encoding semantics from global context in sequences (Vaswani et al., 2017) or images (Caron et al., 2021), we rely on attention to design our Local feature Integration Transformer (*LIT*), a module that outputs an ordered set of *Super-features*.

Let LIT be represented by function $\Phi(\mathcal{U}) : \mathbb{R}^{L \times D} \to \mathbb{R}^{N \times d}$ that takes as input the set of local features $\mathcal{U}$ and outputs $N$ Super-features. We define LIT as an *iterative* module:

$$\Phi(\mathcal{U}) = \mathcal{Q}^T, \qquad \mathcal{Q}^t = \phi(\mathcal{U}; \mathcal{Q}^{t-1}), \tag{3}$$

where $\phi$ denotes the core function of the module applied $T$ times, and $\mathcal{Q}^0 \in \mathbb{R}^{N \times d}$ denotes a set of learnable *templates*, *i.e.* a matrix of learnable parameters. Super-features are progressively formed by iterative refinement of the templates, conditioned on the local features from the CNN.

The architecture of the core function $\phi$ is inspired by the Transformer architecture (Vaswani et al., 2017) and is composed of a dot-product attention function $\psi$, followed by a multi-layer perceptron (MLP). The dot-product attention function $\psi$ receives three inputs, the *key*, the *value* and the *query*[1] which are passed through layer normalization and fed to linear projection functions $K$, $V$ and $Q$ that project them to dimensions $d_k$, $d_v$ and $d_q$, respectively. In practice, we set $d_k{=}d_v{=}d_q{=}d{=}1024$.

The key and value inputs are set as the local features $\boldsymbol{u}_l \in \mathcal{U}$ across all iterations. The query input is the set of templates $\mathcal{Q}^t = \{\boldsymbol{q}_n^t \in \mathbb{R}^d, n = 1 .. N\}$. It is initialized as the learnable templates $\mathcal{Q}^0$ for iteration 0, and is set as the previous output of function $\phi$ for the following iterations. After projecting with the corresponding linear projection functions, the key and the query are multiplied to construct a set of $N$ *attention maps* over the local features, *i.e.*, the *columns* of matrix $\boldsymbol{\alpha} \in \mathbb{R}^{L \times N}$, while the $L$ rows $\boldsymbol{\alpha}_l$ of that matrix can be seen as the *responsibility* that each of the $N$ templates has for each local feature $l$, and is given by[2]:

$$\boldsymbol{\alpha} = \begin{bmatrix} \boldsymbol{\alpha}_1 \\ \vdots \\ \boldsymbol{\alpha}_L \end{bmatrix} \in \mathbb{R}^{L \times N}, \quad \boldsymbol{\alpha}_l = \frac{\hat{\boldsymbol{\alpha}}_l}{\sum_{i=1}^{L} \hat{\boldsymbol{\alpha}}_i}, \quad \hat{\boldsymbol{\alpha}}_l = \frac{e^{M_l}}{\sum_{n=1}^{N} e^{M_{ln}}}, \quad M_{ln} = \frac{K(\boldsymbol{u}_l) \cdot Q(\boldsymbol{q}_n^t)}{\sqrt{d}}. \tag{4}$$

The dot product between keys and templates can be interpreted as a tensor of compatibility scores between local features and templates. These scores are normalized across templates via a softmax function, and are further turned into attention maps by $l_1$ normalization across all $L$ spatial locations. This is a common way of approximating a joint normalization function across rows and columns, also used by Locatello et al. (2020). The input value is first projected with $V$, re-weighted with the attention maps and then residually fed to a `MLP`[3] to produce the output of function $\phi$:

$$\mathcal{Q}^t = \phi(\mathcal{U}; \mathcal{Q}^{t-1}), \quad \phi(\mathcal{U}; \mathcal{Q}) = \texttt{MLP}(\psi(\mathcal{U}; \mathcal{Q})) + \psi(\mathcal{U}; \mathcal{Q}), \quad \psi(\mathcal{U}; \mathcal{Q}) = V(\mathcal{U}) \cdot \boldsymbol{\alpha} + \mathcal{Q}. \tag{5}$$

Following standard image retrieval practice, we further whiten and $l_2$-normalize the output of $\Phi(\mathcal{U})$ to get the final set of Super-features. Specifically, let $\Phi(\mathcal{U}) = [\hat{\boldsymbol{s}}_1; \dots; \hat{\boldsymbol{s}}_N] \in \mathbb{R}^{N \times d}$ be the raw output of our iterative attention module, we define the ordered set of Super-features as:

$$\mathcal{S} = \left\{ \boldsymbol{s}_n : \boldsymbol{s}_n = \frac{o(\hat{\boldsymbol{s}}_n)}{\|o(\hat{\boldsymbol{s}}_n)\|_2}, \quad n = 1, .., N \right\}, \tag{6}$$

where, as in Section 2, $o(\cdot)$ denotes dimensionality reduction and whitening. Figure 2 (right) illustrates the architecture of LIT. Note that all learnable parameters of $\phi$ are shared across all iterations.

**What do Super-features attend to?** Each Super-feature in $\mathcal{S}$ is a function of all local features in $\mathcal{U}$, invariant to permutation of its elements, and can thus attend arbitrary regions of the image.

---

[1] The term 'query' has a precise meaning for retrieval; yet, for this subsection only, we overload the term to refer to one of the inputs of the dot-product attention, consistently with the terminology from seminal works on attention by Vaswani et al. (2017).

[2] The attention maps presented in Eq.(4) are technically taken at iteration $t$, but we omit iteration superscripts for clarity. For the rest of the paper and visualizations, we use *attention maps* to refer to the attention maps of Eq.(4) after the final ($T$-th) iteration of the iterative module.

[3] The `MLP` function consists of a layer-norm, a fully-connected layer with half the dimensions of the features, a ReLU activation and a fully-connected layer that projects features back to their initial dimension.

To visualize the patterns captured by Super-features, Figure 1 shows the attention maps of the same five Super-features for three images, including two of the same landmark (first two rows). The type of patterns depends on the learned initialization templates $\mathcal{Q}^0$; this explains why the Super-features form an ordered set, a property which allows to directly compare Super-features with the same ID. We observe the attention maps to be similar across images of the same landmark and to contain some mid-level patterns (such as a half-circle on the second column, or windows on the third one).

## 3.2 LEARNING WITH SUPER-FEATURES

We jointly fine-tune the parameters of the CNN and of the LIT module using contrastive learning. However, departing from recent approaches like HOW (Tolias et al., 2020) or DELG (Cao et al., 2020) that use a global contrastive loss similar to the one presented in Eq. (2), we introduce a contrastive loss that operates directly on Super-features, *the representations we use at test time*, and yet only requires image-level labels. Given a positive pair (*i.e.* matching images of the same landmark) and a set of negative images, we select promising Super-feature pairs by exploiting their ordered nature and without requiring any extra supervision signal. We then minimize their pairwise distance, while simultaneously reducing the spatial redundancy of Super-features within an image.

**Selecting matching Super-features.** Since we are only provided with pairs of matching images, *i.e.* image-level labels, defining correspondences at the Super-feature level is not trivial. Instead of approximating sophisticated metrics (Liu et al., 2020; Mialon et al., 2021)–see Section 5 for a discussion–, we leverage the fact that Super-features are ordered and we select promising matches and filter out erroneous ones only relying on simple, nearest neighbor-based constraints.

For any $s \in \mathcal{S}$, let $i(s)$ be the function that returns the position/order or *Super-feature ID*, *i.e.* $i(s_i) = i$, $\forall s_i \in \mathcal{S}$. Further let function $n(s, \mathcal{S}) = \arg\min_{s_i \in \mathcal{S}} \|s - s_i\|_2$ be the function that returns the nearest neighbor of $s$ from set $\mathcal{S}$. Now, given a positive pair of images $x, x^+$, let $s \in \mathcal{S}, s' \in \mathcal{S}'$ be two Super-features from their Super-feature sets $\mathcal{S}, \mathcal{S}'$, respectively. In order for Super-feature pair $(s, s')$ to be *eligible*, all of the following criteria must be met: a) $s, s'$ have to be reciprocal nearest neighbors, b) they need to pass Lowe's first-to-second neighbor ratio test (Lowe, 2004) and c) they need to have the same Super-feature ID. Let $\mathcal{P}$ be the set of eligible pairs and $\tau$ the hyper-parameter that controls the ratio test; the conditions above can formally be written as:

$$(s, s') \in \mathcal{P} \iff \begin{cases} s = n(s', \mathcal{S}) \\ s' = n(s, \mathcal{S}') \end{cases} \text{and} \begin{cases} i(s) = i(s') \\ \|s - s'\|_2 \,/\, \|s' - n(s', \mathcal{S}\backslash\{s\})\|_2 \geqslant \tau \end{cases}. \quad (7)$$

We set $\tau = 0.9$. Our ablations (Section 4.1) show that all criteria are important. Note that the pair selection step is non-differentiable and no gradients are computed for Super-features not in $\mathcal{P}$. We further discuss this in Appendix B.4 and empirically show that all Super-features are trained.

**A contrastive loss on Super-features.** Once the set $\mathcal{P}$ of all eligible Super-feature pairs has been constructed, we define a contrastive margin loss on these matches. Let pair $p = (s, s^+) \in \mathcal{P}$ be a pair of Super-features, selected from a pair of matching images $x, x^+$, and let $\mathcal{N}(j) = \{n_j^k : k = 1 .. n\}$ for $j = 1 .. N$ be the set of Super-features with Super-feature ID $j$ extracted from negative images $(y_1^-, \ldots, y_n^-)$. The contrastive Super-feature loss can be written as:

$$\mathcal{L}_{super} = \sum_{(s, s^+) \in \mathcal{P}} \left[ \|s - s^+\|_2^2 + \sum_{n \in \mathcal{N}(i(s))} [\mu' - \|s - n\|_2^2]^+ \right], \quad (8)$$

where $\mu'$ is a margin hyper-parameter and the negatives for each $s$ are the Super-features from all $n$ negative images of the training tuple with Super-feature ID equal to $i(s)$.

**Reducing the spatial correlation between attention maps.** To obtain Super-features that are as complementary as possible, we encourage them to attend to different local features, *i.e.* different locations of the image. To this end, we minimize the cosine similarity between the attention maps of all Super-features of every image.

Specifically, let matrix $\alpha = [\tilde{\alpha}_1, \ldots, \tilde{\alpha}_N]$ now be seen as column vectors denoting the $N$ attention maps after the last iteration of LIT. The *attention decorrelation loss* is given by:

$$\mathcal{L}_{attn}(x) = \frac{1}{N(N-1)} \sum_{i \neq j} \frac{\tilde{\alpha}_i^\top \cdot \tilde{\alpha}_j}{\|\tilde{\alpha}_i\|_2 \|\tilde{\alpha}_j\|_2}, \quad i, j \in \{1, .., N\}. \quad (9)$$

| reci-proc. | ratio test | same ID | SfM-120k val | $\mathcal{R}$Oxford | | $\mathcal{R}$Paris | |
|---|---|---|---|---|---|---|---|
| | | | | med | hard | med | hard |
| | | | 68.3 | 64.3 | 39.8 | 74.1 | 52.4 |
| ✓ | | | 79.6 | 69.2 | 44.3 | 79.2 | 60.9 |
| ✓ | ✓ | | 80.8 | 70.7 | 45.1 | 80.3 | 61.9 |
| | | ✓ | 75.9 | 63.8 | 35.1 | 77.3 | 56.5 |
| ✓ | ✓ | ✓ | **89.7** | **81.8** | **61.2** | **85.3** | **70.0** |

Table 1: **Ablation on matching constraints**: Impact of removing constraints on reciprocity, Lowe's ratio test and the Super-feature ID.

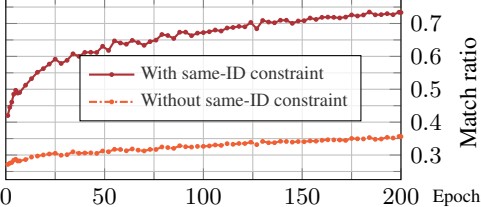

Figure 3: **Evolution of the matching quality** measured as the ratio of matches coming from the positive pair over all pairs with the query in each training tuple.

In other words, this loss minimizes the off-diagonal elements of the $N \times N$ self-correlation matrix of $\tilde{\boldsymbol{\alpha}}$. We ablate the benefit of this loss and others components presented in this section in Section 4.1.

**Image retrieval with Super-features.** Our full pipeline, FIRe, is composed of a model that outputs Super-features, trained with the contrastive Super-feature loss of Eq.(8) and the attention decorrelation loss of Eq.(9). As our ablations show (see Section 4), the combination of these two losses is required for achieving state-of-the-art performance, while adding a third loss on the aggregated global features as in Eq.(2) does not bring any gain.

## 4 EXPERIMENTS

This section validates our proposed FIRe approach on standard landmark retrieval tasks. We use the SfM-120k dataset (Radenović et al., 2018b) following the 551/162 3D model train/val split from Tolias et al. (2020). For testing, we evaluate instance-level search on the $\mathcal{R}$Oxford (Philbin et al., 2007) and the $\mathcal{R}$Paris (Philbin et al., 2008) datasets in their revisited version (Radenović et al., 2018a), with and without the 1 million distractor set called $\mathcal{R}$1M. They both contain 70 queries, with 4993 and 6322 images respectively. We report mean average precision (mAP) on the Medium (med) and Hard (hard) setups, or the average over these two setups (avg).

**Image search with Super-features.** At test time, we follow the exact procedure described by Tolias et al. (2020) and extract Super-features from each image at 7 resolutions/scales {2.0, 1.414, 1.0, 0.707, 0.5, 0.353, 0.25}. We then keep the top Super-features (the top 1000 unless otherwise stated) according to their L2 norm and use the binary version of ASMK with a codebook of 65536 clusters. Note that one can measure the memory footprint of a database image $\boldsymbol{x}$ via the number of non-empty ASMK clusters, *i.e.* clusters with at least one assignment, denoted as $|\mathcal{C}(\boldsymbol{x})|$.

**Implementation details.** We build our codebase on HOW (Tolias et al., 2020)[4] and sample tuples composed of one query image, one positive image and 5 hard negatives. Each epoch is composed of 400 batches of 5 tuples each, while hard negatives are updated at each epoch using the global features of Eq.(1). We train our model for 200 epochs on SfM-120k using an initial learning rate of $3.10^{-5}$ and random flipping as data augmentation. We multiply the learning rate by a factor of 0.99 at each epoch, and use an Adam optimizer with a weight decay of $10^{-4}$. We use a ResNet50 (He et al., 2016) without the last convolutional block as backbone (R50$^-$). For LIT, we use $D = d_k = d_q = d_v = d = 1024$. Following HOW, we reduce the dimensionality of features to 128 and initialize $o(\cdot)$ using PCA and whitening before training and keep it frozen. We pretrain the LIT module with the backbone on ImageNet-1K for image classification, see details in Appendix B.6. We use $\mu' = 1.1$ in Eq.(8) and weight $\mathcal{L}_{super}$ and $\mathcal{L}_{attn}$ with 0.02 and 0.1, respectively.

### 4.1 ANALYSIS AND ABLATIONS

In this section, we analyze the proposed FIRe framework and perform exhaustive ablations on the training losses and the matching constraints. The impact of the number of iterations ($T$) and templates ($N$) in LIT is studied in Appendix A.1. For the rest of the paper, we set $T = 6$ and $N = 256$.

**Matching constraints ablation.** Table 1 reports the impact of the different matching constraints: nearest neighbor, reciprocity, Lowe ratio test and constraint on Super-features ID. Adding reciprocity

---

[4]https://github.com/gtolias/how

| $\mathcal{L}_{global}$ | $\mathcal{L}_{attn}$ | $\mathcal{L}_{super}$ | SfM-120k val | $\mathcal{R}$Oxford med | hard | $\mathcal{R}$Paris med | hard |
|:---:|:---:|:---:|:---:|:---:|:---:|:---:|:---:|
| ✓ | | | 79.0 | 64.3 | 38.0 | 75.4 | 51.7 |
| ✓ | ✓ | | 87.7 | 75.8 | 51.2 | 79.0 | 57.0 |
| ✓ | ✓ | ✓ | 88.4 | 79.0 | 57.2 | 83.0 | 65.6 |
| | | ✓ | 61.7 | 59.3 | 32.8 | 69.9 | 47.3 |
| | ✓ | ✓ | **89.7** | **81.9** | **61.5** | **85.3** | **70.1** |

Table 2: **Ablation on loss components**: Impact of removing $\mathcal{L}_{attn}$, using either a global loss $\mathcal{L}_{global}$ or a loss directly on Super-features $\mathcal{L}_{super}$ or a combination of both.

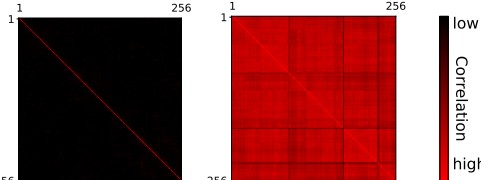

Figure 4: **Impact of $\mathcal{L}_{attn}$ on the correlation matrix between attention maps** (the darker, the lower is the correlation) at the last iteration of LIT when training with (left) and without (right). This is averaged over the 70 queries of $\mathcal{R}$Oxford.

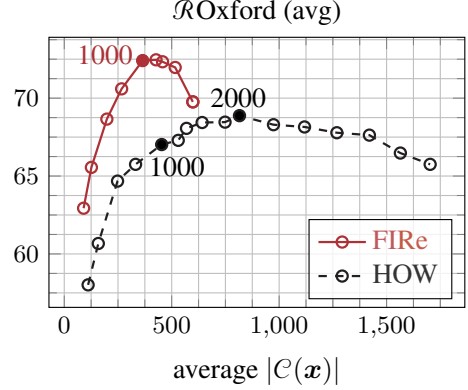 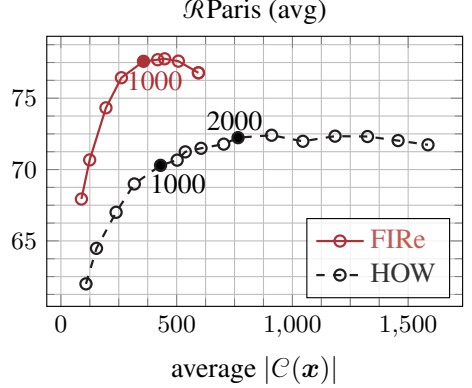

Figure 5: **Performance versus memory** for HOW and FIRe. The x-axis shows the average number of clusters per image used in ASMK (proportional to memory usage). We vary the number of features extracted per image before aggregation in $\{200, \ldots, 2000, 2500, \ldots, 5000\}$; solid markers denote the commonly used settings (1000/2000). FIRe has at most 1,792 features (256 per scale).

and Lowe's ratio test significantly improves performance, which indicates that it reduces the number of incorrect matches. Keeping all pairs with the same ID yields lower performance: the ID constraint alone is not enough. One possible explanation is that two images from the same landmark may differ significantly, *e.g.* when some parts are visible in only one of the two images. In that case, the other constraints allow features attending non-overlapping regions of a positive image pair to be excluded from $\mathcal{P}$. Finally, combining the selective ID constraint with the others yields the best performance.

In order to better understand the impact of the Super-feature ID constraint, a constraint only applicable for Super-features due to their ordered nature, we measure the quality of selected matches during training with and without it. Since there is no ground-truth for such localized matches on landmark retrieval datasets, we measure instead the ratio of matches coming from the positive pair, over all matches (from the positive and all negatives). Ideally, a minimal number of matches should come from the negatives, hence this ratio should be close to 1. In Figure 3 we plot this match ratio for all epochs; we observe that it is significantly higher when using the Super-features ID constraint.

**Training losses ablation.** We study the impact of the different training losses in Table 2. We start from a global loss similar to HOW (first row). We then add the decorrelation loss (second row) and observe a clear gain. It can be explained by the fact that without this loss, Super-features tend to be redundant (see the correlation matrices in Figure 4). Adding the loss operating directly on the Super-features further improves performance (third row). Next, we remove the global loss and keep only the loss on Super-features alone (fourth row) or with the decorrelation loss (last row). The latter performs best (fourth vs last row). Figure 4 displays the correlation matrix of Super-features attention maps with and without $\mathcal{L}_{attn}$. Without it, we observe that most Super-features have correlated attentions. In contrast, training with $\mathcal{L}_{attn}$ leads to uncorrelated attention maps. This is illustrated in Figure 1 which shows several attention maps focusing on different areas.

**Varying the number of features at test time.** Figure 5 compares HOW (Tolias et al., 2020) with our approach, as we vary the number of local features / Super-features. The x-axis shows the average number of clusters used in ASMK for the database images, *i.e.*, which is proportional to the average memory footprint of an image. We observe that our approach does not only significantly improve

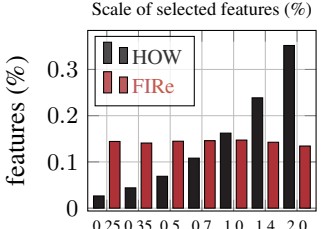 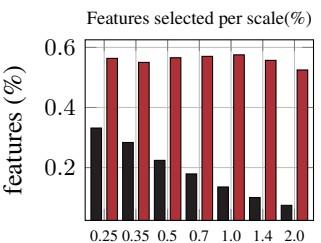 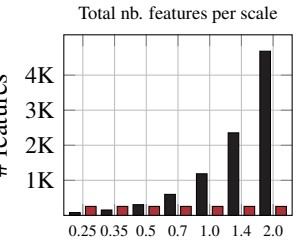

Figure 6: **Statistics on the feature selected across scales** for HOW and FIRe, averaged over the 70 queries from $\mathcal{R}$Oxford. *Left:* Among the 1000 selected features, we show the percentage coming from each scale. *Middle:* For each scale, we show the ratio of features that are selected. *Right:* Total number of features per scale; we extract $N = 256$ Super-features regardless of scale.

accuracy compared to HOW, but also requires overall less memory. Notably, FIRe matches the best performance of HOW with a memory footprint reduced by a factor of 4. For both methods, performance goes down once feature selection no longer discards background features. The gain in terms of performance is even larger when considering a single scale at test time (see Appendix B.1).

**Statistics on the number of selected features per scale.** The left plot of Figure 6 shows the percentage of the 1000 selected features that comes from each scale, for HOW and FIRe. For HOW most selected features come from the higher resolution inputs; by contrast, selected Super-features are almost uniformly distributed across scales. Interestingly, Figure 6 (middle) shows that HOW keeps a higher percentage of the features coming from coarser scales. Yet, the final feature selection for HOW is still dominated by features from higher scales, due to the fact that the number of local features significantly increases with the input resolution (see Figure 6 right).

## 4.2 COMPARISON TO THE STATE OF THE ART

We compare our method to the state of the art in Table 3. All reported methods are trained on SfM-120k for a fair comparison.[5] First, we observe that methods based on global descriptors (Tolias et al., 2016; Revaud et al., 2019a) compared with a L2 distance tend to be less robust than methods based on local features (Noh et al., 2017; Tolias et al., 2020). DELG (Cao et al., 2020) shows better performance, owing to a re-ranking step based on local features, at the cost of re-extracting local features for the top-ranked images of a given query, given that local features would take too much memory to store. Our FIRe method outperforms DELF (Noh et al., 2017) as well as HOW (Tolias et al., 2020) by a significant margin when extracting 1000 features per image. Importantly, our approach also requires less memory, as it uses fewer ASMK clusters per image, as shown in Figure 5: for the whole $\mathcal{R}$1M set, HOW uses 469 clusters per image while FIRe uses only 383, on average.

## 5 RELATED WORK

**Image descriptors for retrieval.** The first approaches for image retrieval were based on handcrafted local descriptors and bag-of-words representations borrowed from text retrieval (Sivic & Zisserman, 2003; Csurka et al., 2004), or other aggregation techniques like Fisher Vectors (Perronnin et al., 2010), VLAD (Jégou et al., 2010) or ASMK (Tolias et al., 2013). First deep learning techniques were extracting a global vector per image either directly or by aggregating local activations (Gordo et al., 2016; Radenović et al., 2018b) and have shown to highly outperform handcrafted local features, see (Csurka & Humenberger, 2018) for a review. Methods that perform matching or re-ranking using CNN-based local features are currently the state of the art in the area (Noh et al., 2017; Teichmann et al., 2019; Cao et al., 2020; Tolias et al., 2020). They are able to learn with global (*i.e.* image-level) annotations. Most of them use simple variants of attention mechanisms (Noh et al., 2017; Ng et al., 2020) or simply the feature norm (Tolias et al., 2020) to weight local activations.

**Low-level features aggregation.** Aggregating local features into regional features has a long history. Crucial to the success of traditional approaches, popular methods include selecting discriminative patches in the input image (Singh et al., 2012; Gordo, 2015), regressing saliency map scores

---

[5]The GLDv2-clean dataset (Yokoo et al., 2020) is sometimes used for training. Yet, there is significant overlap between its training classes and the $\mathcal{R}$Oxford and $\mathcal{R}$Paris query landmarks. This departs from standard image retrieval evaluation protocols. See Appendix B.8 for details.

| method | FCN | Mem (GB) | $\mathcal{R}$Oxford | | $\mathcal{R}$Oxford +$\mathcal{R}$1M | | $\mathcal{R}$Paris | | $\mathcal{R}$Paris +$\mathcal{R}$1M | |
|---|---|---|---|---|---|---|---|---|---|---|
| | | | med | hard | med | hard | med | hard | med | hard |
| *Global descriptors* | | | | | | | | | | |
| RMAC (Tolias et al., 2016) | R101 | 7.6 | 60.9 | 32.4 | 39.3 | 12.5 | 78.9 | 59.4 | 54.8 | 28.0 |
| AP-GeM‡ (Revaud et al., 2019a) | R101 | 7.6 | 67.1 | 42.3 | 47.8 | 22.5 | 80.3 | 60.9 | 51.9 | 24.6 |
| GeM+SOLAR (Ng et al., 2020) | R101 | 7.6 | 69.9 | 47.9 | 53.5 | 29.9 | 81.6 | 64.5 | 59.2 | 33.4 |
| *Global descriptors + reranking with local features* | | | | | | | | | | |
| DELG (Cao et al., 2020) | R50 | 7.6 | 75.1 | 54.2 | 61.1 | 36.8 | 82.3 | 64.9 | 60.5 | 34.8 |
| DELG (Cao et al., 2020) | R101 | 7.6 | *78.5* | *59.3* | *62.7* | *39.3* | *82.9* | *65.5* | *62.6* | *37.0* |
| *Local features + ASMK matching (max. 1000 features per image)* | | | | | | | | | | |
| DELF (Noh et al., 2017) | R50⁻ | 9.2 | 67.8 | 43.1 | 53.8 | 31.2 | 76.9 | 55.4 | 57.3 | 26.4 |
| DELF-R-ASMK (Teichmann et al., 2019) | R50⁻ | 27.4 | 76.0 | 52.4 | 64.0 | 38.1 | 80.2 | 58.6 | 59.7 | 29.4 |
| HOW (Tolias et al., 2020) | R50⁻ | 7.9 | 78.3 | 55.8 | 63.6 | 36.8 | 80.1 | 60.1 | 58.4 | 30.7 |
| **FIRe** (ours) | R50⁻ | **6.4** | **81.8** | **61.2** | **66.5** | **40.1** | **85.3** | **70.0** | **67.6** | **42.9** |
| *(standard deviation over 5 runs)* | | | (±0.6) | (±1.0) | (±0.8) | (±1.1) | (±0.4) | (±0.6) | (±0.7) | (±0.8) |
| *(mAP gains over HOW)* | | | (↑ **3.5**) | (↑ **5.4**) | (↑ **2.9**) | (↑ **3.3**) | (↑ **5.2**) | (↑ **9.9**) | (↑ **9.2**) | (↑ **12.2**) |

Table 3: **Comparison to the state of the art.** All models are trained on SfM-120k. FCN denotes the fully-convolutional network backbone, with R50⁻ denoting a ResNet-50 without the last block. Memory is reported for the image representation of the full $\mathcal{R}$1M set (without counting local features for the global descriptors + reranking methods). ‡ result from (Tolias et al., 2020). **Bold** denotes best performance, underlined second best among methods using ASMK, *italics* second best overall.

at different resolutions (Jiang et al., 2013), aggregating SIFT descriptors with coding and pooling schemes (Boureau et al., 2010), or mining frequent patterns in sets of SIFT features (Singh et al., 2012). More recently, Hausler et al. (2021) introduced a multi-scale fusion of patch features.

**Supervision for local features.** Several works provide supervision at the level of local features in the context of contrastive learning. Xie et al. (2021) and Chen et al. (2021) obtain several views of an input image using data augmentations with known pixel displacements. Similarly, Liu et al. (2020) train a model to predict the probability for a pair of local features to match, evaluated using known displacements. Wang et al. (2020) and Zhou et al. (2021) obtain local supervision by relying on epipolar coordinates and relative camera poses. Positive pairs in image retrieval depart from these setups, as pixel-level correspondences cannot be known. To build matches, Wang et al. (2021) use a standard nearest neighbor algorithm to build pairs of features, similarly to our approach, but without the use of filtering which is critical to our final performance. Using the current model predictions to define targets is reminiscent of modern self-supervised learning approaches which learn without any label (Caron et al., 2021; Grill et al., 2020). The additional filtering step can be seen as a way to keep only the most reliable model predictions, similar to *self-distillation* as in *e.g.* Sohn et al. (2020).

**Attention modules.** Our LIT module is an iterative variant of standard attention (Bahdanau et al., 2015; Vaswani et al., 2017), adapted to map a variable number of input features to an ordered set of $N$ output features, similar to Lee et al. (2019). The Perceiver model (Jaegle et al., 2021) has demonstrated the flexibility of such constructions by using it to scale attention-based deep networks to large inputs. Our design was heavily inspired by *slot-attention* (Locatello et al., 2020), but has some key differences that enable us to achieve high performance in more complex visual environments: a) unlike the slot attention which initializes its slots with i.i.d sampling, we *learn* the initial templates and therefore define an ordering on the output set, a crucial property for selecting promising matches; b) we replace the recurrent network gates with a residual connection across iterations. These modifications, together with the attention decorrelation loss enable our module to go from a handful of object-oriented slots to a much larger set of output features. For object detection, Carion et al. (2020) rely on a set of learned *object queries* to initialize a stack of transformer layers. Unlike ours, their module is not recurrent; Appendix A.1 experimentally shows substantial benefits from applying LIT $T$ times, with weight sharing, to increase model flexibility without extra parameters.

## 6 CONCLUSIONS

We present an approach that aggregates local features into Super-features for image retrieval, a task that has up to now been dominated by approaches that work at the local feature level. We design an attention mechanism that outputs an ordered set of such features that are more discriminative and expressive than local features. Exploiting their ordered nature and without any extra supervision, we present a loss working directly on the proposed features. Our method not only significantly improves performance, but also requires less memory, a crucial requirement for scalable image retrieval.

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

# Appendix

## Table of Contents

In this appendix, we present additional ablations (Appendix A), a deeper analysis of our framework (Appendix B) as well as results of the application of our FIRe framework for image retrieval to the task of visual localization (Appendix C). We briefly summarize the findings in the following paragraphs.

**Ablations.** We study the impact of some hyper-parameters of our model, namely the size $N$ of the set of Super-features, and the number of iterations $T$ in the LIT module, in Appendix A.1. We show that 256 Super-features and 6 iterations offer the best trade-off between performance and computational cost. We also show in Appendix A.2 that we obtain further performance gains of over 1% by increasing the number of negatives to 10 or 15. We then study the impact of replacing the residual connection inside the LIT module by a recurrent network (Appendix A.3). In Appendix A.4 we present an extended version of Table 1 with further matching constraints and we finally study the impact of the template initialization on performance in Appendix A.5.

**Properties of Super-features.** We show single-scale results in Appendix B.1. In Appendix B.2, we display the attention maps of Super-features, at different scales for a fixed Super-feature ID. We further study the amount of redundancy in Super-features, compared to local features, in Appendix B.3. Next, we verify in Appendix B.4 that all Super-features receive training signal, as a sanity check. We discuss the case of applying a loss directly on local features in Appendix B.5 and give details about the pretraining on ImageNet in Appendix B.6. In Appendix B.7 we report the average extraction time for the proposed Super-features, while in Appendix B.8 we discuss the fact that there is an overlap between the queries from the common $\mathcal{R}$Oxford and $\mathcal{R}$Paris datasets and the Google Landmarks-v2-clean dataset that is commonly used as training set for retrieval on $\mathcal{R}$Oxford and $\mathcal{R}$Paris.

**Application to visual localization.** We further evaluate our model using a visual localization setting on the Aachen Day-Night v1.1 dataset (Sattler et al., 2018) in Appendix C. To do so, we leverage a retrieval + local feature matching pipeline, and show that it is beneficial to use our method, especially in hard settings.

# A ADDITIONAL ABLATIONS

## A.1 IMPACT OF THE ITERATION AND THE NUMBER OF TEMPLATES HYPER-PARAMETERS

In this section, we perform ablations on the number of Super-features $N$ and the number of iterations $T$ in the Local feature Integration Transformer. We first study the impact of the number $N$ of Super-features extracted for each scale of each image in Figure 7a. We observe that the best value is 256 on both the validation and test sets ($\mathcal{R}$Oxford and $\mathcal{R}$Paris). We then study the impact of the number of iterations T in Figure 7b. While the performance decreases when doing 2 or 3 iterations compared to just 1, a better performance is reached for 6 iterations, after which the performance saturates while requiring more computations. We thus use $T = 6$.

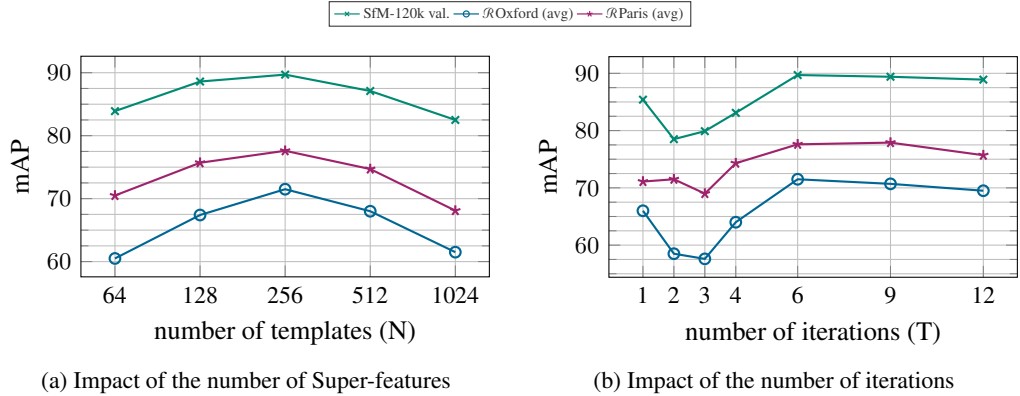

(a) Impact of the number of Super-features     (b) Impact of the number of iterations

Figure 7: **Varying the number of Super-features $N$ and of iterations $T$** in the Local feature Integration Transformer.

## A.2 IMPACT OF HARD NEGATIVES

Each training tuple is composed of one image pair depicting the same landmark, and 5 negative images (*i.e.* from different landmarks). We plot the performance when varying this number in Figure 8. We observe that adding more negatives improves overall the performance on all datasets, *i.e.* by more than 1% when increasing it to 10 or 15 negatives, but at the cost of a longer training time as more images need to be processed at each iteration. This is why we use 5 negatives in the rest of the paper as it offers a good compromise between performance and cost.

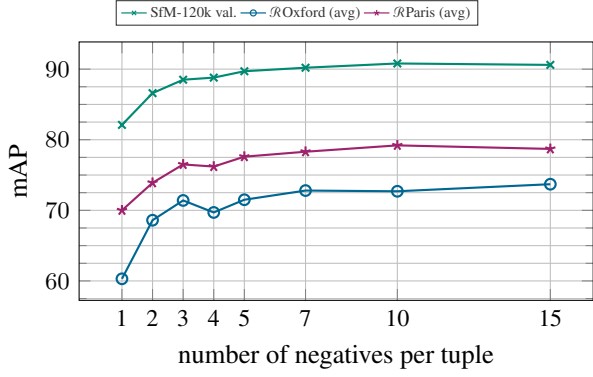

Figure 8: **Varying the number of hard negatives per training tuple.**

### A.3 IMPACT OF THE UPDATE FUNCTION

The formula for $\psi$ in Equation (5) sums the previous $\mathcal{Q}$ with the output of the attention component $V(\mathcal{U}) \cdot \boldsymbol{\alpha}$, *i.e.*, with a residual connection. This is a different choice than the one made in the object-centric slot attention of Locatello et al. (2020), which proposes to use a Gated Recurrent Unit: $\psi(\mathcal{U}; \mathcal{Q}) = \mathrm{GRU}(V(\mathcal{U}) \cdot \boldsymbol{\alpha}, \mathcal{Q})$. We thus compare the residual connection we use to a `GRU` function and report results in Table 4 with $T = 3$. We observe that the residual connection reaches a better performance in all datasets while having the interest of not adding extra parameters.

| update | SfM-120k | $\mathcal{R}$Oxford | | $\mathcal{R}$Paris | |
|--------|----------|------|------|------|------|
| function | val | med | hard | med | hard |
| residual | **79.9** | **71.3** | **43.9** | **78.7** | **59.4** |
| GRU | 74.7 | 66.7 | 41.4 | 77.2 | 57.8 |

Table 4: **Impact of the update function** in the LIT module. We compare the performance of the residual combination of the previous template value with the cross-attention tensor compared to a GRU as used in slot attention (Locatello et al., 2020). In this experiment, we use $T = 3$.

In summary, we propose the LIT module to obtain a few hundred features for image retrieval while slot attention handles a handful of object attentions. Technical differences to the slot attention include: a) we use a learned initialization of the templates instead of i.i.d. sampling, which allows to obtain an ordered set of features, and thus to apply the constraint on the ID for the matching, leading to a clear gain, see Table 1 and Figure 3, b) to handle a larger number of templates, we also add a decorrelation loss on the attention maps, which has clear benefit, see Table 2 and Figure 4, c) we use a residual connection with 6 iterations instead of a `GRU`, leading to improved performance for our task (see Table 4).

### A.4 EXTENDED ABLATION ON MATCHING CONSTRAINTS

| reci-proc. | ratio test | same ID | SfM-120k val | $\mathcal{R}$Oxford med | hard | $\mathcal{R}$Paris med | hard |
|------|------|------|------|------|------|------|------|
| | | | 68.3 | 64.3 | 39.8 | 74.1 | 52.4 |
| ✓ | | | 79.6 | 69.2 | 44.3 | 79.2 | 60.9 |
| ✓ | | ✓ | **89.9** | **81.9** | 61.1 | 85.1 | 69.6 |
| | ✓ | | 73.5 | 64.6 | 39.0 | 75.5 | 55.0 |
| | ✓ | ✓ | 84.2 | 75.0 | 49.8 | 79.4 | 61.1 |
| ✓ | ✓ | | 80.8 | 70.7 | 45.1 | 80.3 | 61.9 |
| | | ✓ | 75.9 | 63.8 | 35.1 | 77.3 | 56.5 |
| ✓ | ✓ | ✓ | 89.7 | 81.8 | **61.2** | **85.3** | **70.0** |

Table 5: **Extended ablation on matching constraints.** We study the impact of removing constraints on reciprocity, Lowe's ratio test and the Super-feature ID.

We show in Table 5 an extended version of Table 1 of the main paper, where we evaluate all possible combinations of constraints among reciprocity, Lowe's ratio test and the Super-feature ID. We observe that the reciprocity constraint and the Super-feature ID constraint are the two most important ones, while the Lowe's ratio only only brings a small improvement.

### A.5 IMPACT OF THE TEMPLATE INITIALIZATION

In the LIT module, initial templates $\mathcal{Q}^0 \in \mathbb{R}^{N \times d}$ are learned together with the LIT module. We can therefore assume that they are adapted to the task at hand. To explore the sensitivity to the initial templates, we run a variant of FIRe where the initializations are not fine-tuned, but instead frozen to the values after pretraining on ImageNet. We report performance in Table 6. We observe that the two variants perform overall similarly. This ablation suggests that the initial templates are up to some point transferable to other tasks.

| Template Initialization | SfM-120k val | $\mathcal{R}$Oxford med | hard | $\mathcal{R}$Paris med | hard |
|---|---|---|---|---|---|
| Frozen from ImageNet pretraining | 89.5 | 81.3 | 59.6 | **85.3** | **70.2** |
| Fine-tuned for landmark retrieval | **89.7** | **81.8** | **61.2** | **85.3** | 70.0 |

Table 6: **Fine-tuning the initial templates.** Comparison where we either fine-tune the initial templates $\mathcal{Q}^0$ of LIT (bottom row, as in the main paper) or keep them frozen after ImageNet pretraining (top row).

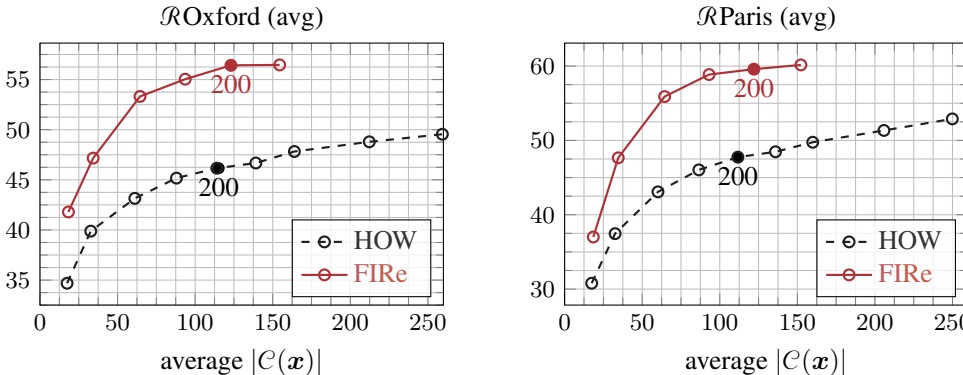

Figure 9: **Performance versus memory when varying the number of selected features at a single scale** for HOW and FIRe. The x-axis represents the average number of vectors per image in ASMK, which is proportional to the memory, when varying the number of selected features in $(25, 50, 100, 150, 200, 300, 400, 500, 600)$, FIRe is limited to 256 features.

## B ANALYSIS AND DISCUSSIONS

### B.1 SINGLE-SCALE RESULTS

Similar to Figure 5 of the main paper, we perform the same ablation in Figure 9 when extracting features at a single scale (1.0). We observe that FIRe significantly improves the mAP on the two datasets compared to HOW. The average number of clusters, *i.e.* the memory footprint of images, remains similar for both methods at a same number of selected features. This stands in contrast to the multi-scale case where our approach allows to save memory (about 20%), which we hypothesize is due to the correlation of our features across scales that we discuss below.

### B.2 CONSISTENCY OF ATTENTION MAPS ACROSS SCALES

At test time, we extract Super-features at different image resolutions. We propose here to study if the attention maps of the Super-features across different image scales are correlated. We show in Figure 10 the attention maps at the last iteration of LIT for the different image scales. We observe that they fire at the same image location at the different scales. Note that the attention maps are larger/smoother at small scales (right columns), as for visualization, we resize lower resolution attention maps to the original image size.

### B.3 REDUNDANCY IN SUPER-FEATURES

To evaluate the redundancy of Super-features versus local features, Figure 11 displays the average cosine similarity between every local feature / Super-feature and its $K$ nearest local features / Super-features from the same image, for different values of $K$. We observe that Super-features are significantly less correlated to the most similar other ones, compared to local features used in HOW.

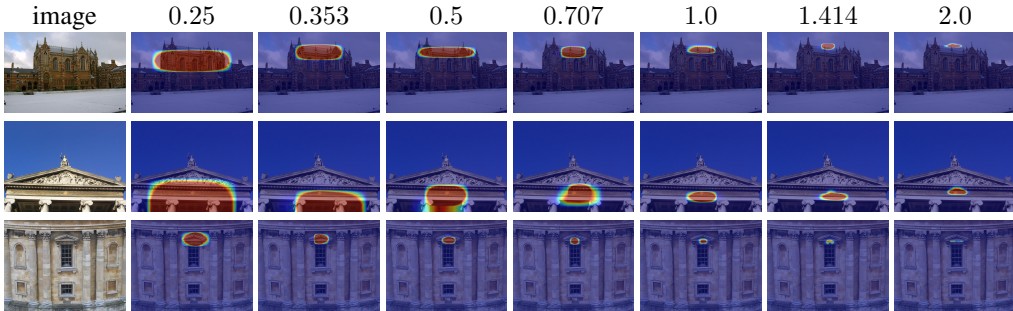

Figure 10: **Attention consistency across scales.** For three images, we select one Super-feature per image and show the attention maps of the latest iteration of LIT for different image scales, with from left to right $0.25, 0.353, 0.5, 0.707, 1.0, 1.414$ and $2.0$. We clearly observe that attention maps are correlated. They show larger regions at small scales as for visualization, we resize the lower resolution attention to the original image size.

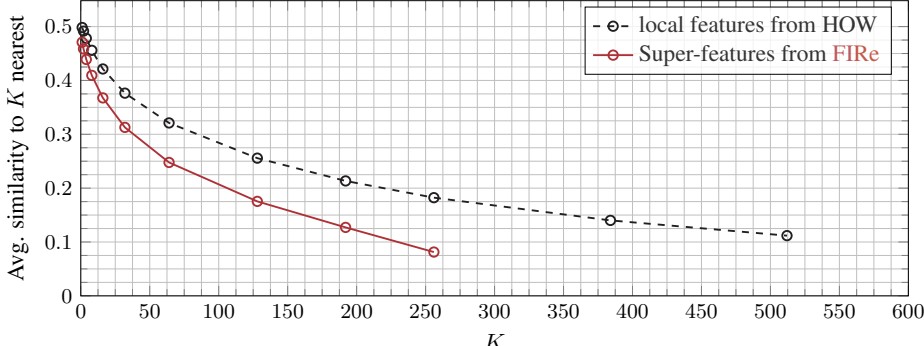

Figure 11: **Measuring Super-feature redundancy.** We compute the average cosine similarity between every feature (local feature from HOW or Super-feature from FIRe) and its K most similar features from the same image, for varying K. Results are averaged over the 70 query images of the $\mathcal{R}$Oxford dataset. Only 256 Super-features can be extracted per image, explaining its maximum x value.

### B.4 ARE ALL SUPER-FEATURES TRAINED?

The loss in Equation (8) only operates on a subset of the Super-feature pairs that pass the criteria of Equation (7); if a Super-feature ID is not matched, it receives no training signal. To investigate if all of the IDs contribute to the loss, we monitored how many times each ID is matched at each epoch and report the percentage over all 2000 training tuples per epoch. In Figure 12 we report mean, standard deviation and minimum number of matches for the $N$ Super-feature ID. We clearly observe that all Super-features receive training signals regularly and each Super-feature is matched about one quarter of the time on average.

### B.5 USING THE SUPER-FEATURE LOSS WITH LOCAL FEATURES

It is worth noting that the loss in Equation (8) could also theoretically be used over local feature activations. It could be computed on pairs of local features that pass the conditions described in Equation (7), after removing the constraint on having the same Super-feature ID. Unfortunately, we were unable to get any gains over HOW (Tolias et al., 2020) when appending such a loss side-by-side with the global loss over local features. Empirically, our ablations (see Section 4.1) show that adding the constraint on the Super-features ID, which is only possible with ordered feature sets, is key to the success of our approach, and significantly improves the quality of the matching.

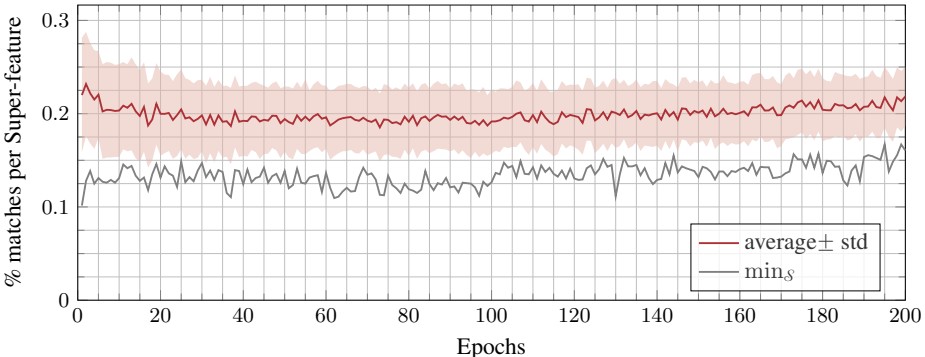

Figure 12: **Are all Super-features trained?** As a sanity check, this plot shows the average number (in percentage) of times (with standard deviation) that the Super-features receive training signal, across training epochs. We also plot the minimum value, which is always significantly positive, showing that each Super-feature ID receives training signal. Thus our loss $\mathcal{L}_{\text{super}}$ proposed in Equation (8) does not lead to degenerate solutions where some IDs are never selected and never trained.

### B.6  Pretraining the backbone together with LIT

Before training on landmark retrieval, we pretrain the backbone network, with the proposed LIT module appended after the last layer, for image classification on ImageNet-1K. Given that we remove the last convolutional block similar to HOW, to avoid overfitting we further append a classification head composed of one fully-connected layer, batch norm, leaky ReLU, dropout and another fully-connected layer at the end of the network and train it for 80 epochs using a standard cross-entropy loss and in addition the decorrelation loss $\mathcal{L}_{\text{attn}}$ weighted by $0.1$, the same weight as during fine-tuning. We start with a learning rate of $0.01$ and divide it by 10 every 20 epochs. This new architecture reaches a top-1 (resp. top-5) accuracy of $73.48\%$ (resp. $91.54\%$) on the ImageNet validation set. Note that this performance is a bit lower than a standard ResNet-50: this is because we have removed the last convolutional block which contains most parameters in the ResNet architecture.

### B.7  Computational cost of Super-features extraction

We report the time required for extracting multi-scale features for 5000 images, for HOW and FIRe. On our server, it took 157 seconds for HOW and 172 for FIRe, *i.e.* extraction for Super-features only requires 10% more wall-clock time.

### B.8  Is Google Landmarks v2 clean an appropriate training dataset for testing on $\mathcal{R}$Oxford and $\mathcal{R}$Paris?

In the comparison to the state of the art (Table 3), all reported methods are trained on the SfM-120k dataset. Several recent works have also released a model trained on the Google Landmarks v2 dataset (Weyand et al., 2020) or its clean version (Yokoo et al., 2020) with excellent performance on $\mathcal{R}$Paris. However, we find out that several of the *query landmarks* from $\mathcal{R}$Paris and $\mathcal{R}$Oxford were present in the training set of the cleaned version of Google Landmarks v2, such as 'La Defense', 'Eiffel Tower', 'Sacré Coeur' or 'Hotel les Invalides' for $\mathcal{R}$Paris or such as 'Mary Madgalen', 'Bodleian Library', 'All Souls College' or 'Radcliffe' for $\mathcal{R}$Oxford.

## C  Application to visual localization

In this section, we evaluate FIRe for the task of visual localization, where retrieval is used as a first-stage filtering and before more precise, local feature-based geometric matching. To this end, we follow the pipeline proposed by Kapture[6] (Humenberger et al., 2020) on the Aachen Day-Night v1.1 dataset (Sattler et al., 2018). In this scenario, a global Structure-from-Motion map is built

---

[6]https://github.com/naver/kapture-localization

| Retrieval method | Day images | | | Night images | | |
|---|---|---|---|---|---|---|
| | 0.25m, 2° | 0.5m, 5° | 5m, 10° | 0.25m, 2° | 0.5m, 5° | 5m, 10° |
| AP-GeM (Revaud et al., 2019a) | 88.8 | **96.6** | **99.6** | 72.3 | 86.9 | 97.9 |
| HOW (Tolias et al., 2020) | **90.8** | 96.2 | **99.6** | 72.8 | **90.1** | 97.9 |
| **FIRe (ours)** | 90.7 | 96.5 | 99.5 | **74.3** | **90.1** | **98.4** |

Table 7: **Visual localization results.** Percentage of successfully localized images on the Aachen Day-Night v1.1 dataset when changing the retrieval method in the Kapture pipeline from Humenberger et al. (2020). In the most challenging scenario, *i.e.* night images at strictest localization threshold, using FIRe yields a 2% improvement compared to AP-GeM and a 1.5% improvement compared to HOW. **Bold** number denotes the best performance, underlined indicates performance within a 0.1 margin to the best one.

from the training images using R2D2 local descriptors (Revaud et al., 2019b). At test time, given a query image to localize, image retrieval is used to retrieve the top-50 nearest images. On these retrieved images, R2D2 local features are extracted and matched with the ones on the query image, and this is used to estimate the position of the camera. The percentage of successfully localized images within three levels of thresholds is then reported on the day or night images, following the visual localization benchmark protocol[7], the latter ones being more challenging as training images are taken during daytime.

We compare our retrieval method to AP-GeM (Revaud et al., 2019a) and HOW (Tolias et al., 2020) and report results in Table 7. AP-GeM is the default method used in Kapture (Humenberger et al., 2020). We observe that using FIRe leads to better visual localization, specially in the most challenging scenario of night image localization and the strictest localization threshold: Performance improves by 2% compared to AP-GeM and by 1.5% compared to HOW on night images at a threshold of 0.25m and 2°. Overall, for either day or night images, FIRe either improves or performs on par to both methods compared.

---

[7]https://www.visuallocalization.net/

