# OpenReview forum: "Learning Super-Features for Image Retrieval"
_ICLR.cc/2022/Conference — ICLR 2022 Poster_

### Official Review · Reviewer_9CXB · 2021-10-23

**Correctness:** 4
**Technical Novelty And Significance:** 4
**Empirical Novelty And Significance:** 3
**Recommendation:** 8
**Confidence:** 4

**Details Of Ethics Concerns:**

No comments.

**Main Review:**

Strengths:

1) A new method called Feature Integration-based Retrieval or FIRe for short.
2) Paper is well written.
3) A comparisson with other state-of-the-art methods. Authors show that their methods permers better on selected databases.
4) Reference section is updated.

Weaknesses:

Not particularly found.

**Summary Of The Paper:**

Authors describe an architecture for deep image retrieval. For this, they use mid-level features that they call super-features (SF). They construct thes SF by an iterative attention module. They build them as an ordered set in which each element focuses on a localized yet discriminant image pattern. Auhtors show that for training, only image labels suffice. Authors present a set of experiments on common landmark retrieval benchmarks to validate that their SF substantially outperform state-of-the-art methods when using the same number of features. They also show that this requires a significantly smaller memory footprint to match their performance.

**Summary Of The Review:**

Authors describe an architecture for deep image retrieval. For this, they use mid-level features that they call super-features (SF). They construct thes SF by an iterative attention module. They build them as an ordered set in which each element focuses on a localized yet discriminant image pattern. Auhtors show that for training, only image labels suffice. Authors present a set of experiments on common landmark retrieval benchmarks to validate that their SF substantially outperform state-of-the-art methods when using the same number of features. They also show that this requires a significantly smaller memory footprint to match their performance.

Main contribution: A method called: Feature Integration-based Retrieval or FIRe for short. It can decomposed into 3 main parts:

1) an image representation based on mid-level features (super-features) and an iterative module to extract them.
2) a framework to learn such representations. It is based on a loss applied directly on SF yet only requiring image-level labels.
3) a set of extensive evaluations that show significant performance gains over the state of the art for landmark image retrieval.

Some possible works that could be added to reference section:

Intellige: A User-Facing Model Explainer for Narrative Explanations
J Yang, D Negoescu, P Ahammad - arXiv preprint arXiv:2105.12941, 2021 - arxiv.org

Next generation 3D pharmacophore modeling
D Schaller, D Šribar, T Noonan, L Deng… - Wiley …, 2020 - Wiley Online Library

Automated cyberbullying detection in social media using an svm activated stacked convolution lstm network
TA Buan, R Ramachandra - Proceedings of the 2020 the 4th …, 2020 - dl.acm.org

AUTOMATIC MODULATION CLASSIFIER
HA Rasool - Iraqi Journal of Information & Communications …, 2020 - ijict.edu.iq

Next generation 3D pharmacophore modeling
D Šribar, T Noonan, L Deng… - WIREs …, 2020 - refubium.fu-berlin.de

Synergistic Target Combinations Against Obesity: Focus on MCHR1/H3R Modulation
DA Schaller - 2020 - refubium.fu-berlin.de

---

> ### Author Response · Authors · 2021-11-17
> **Author's Response**
>
> We would like to thank the reviewer for an overall positive evaluation and general appreciation of our work.
>
> We also thank the reviewer for suggesting papers to be potentially added to the related work section. Yet, we did not manage to find any direct connection between those and the method proposed in our paper. We would welcome further discussion on the rationale for suggesting those prior works.

---

### Official Review · Reviewer_hp4Y · 2021-10-29

**Correctness:** 4
**Technical Novelty And Significance:** 3
**Empirical Novelty And Significance:** 3
**Recommendation:** 8
**Confidence:** 4

**Main Review:**

+ The experimental results are solid and relevant. The authors' approach outperform the baselines by a large margin, not only in mAP but also in its memory requirements. The ablation studies are complete and contribute to understanding the role of each part of the approach. This reviewer finds very relevant the fact that a global loss is irrelevant, as it is the dominant approach in most papers in the literature.
+ Although it is not its strongest point, several details on the architecture and the losses are novel. This might be a critical point in a machine learning conference as ICLR: there are a few small contributions but not groundbreaking advances in this paper. In the opinion of this reviewer, the novel contributions plus the significantly improved performance is enough for acceptance.
+ The research is excellently written and presented.
- I miss Patch-NetVLAD [Hausler et al. CVPR21] in the list of references and the baselines. I am not expecting it to outperform the authors' approach, as Hausler et al. report worse performance than DELG, that is outperformed by this research. But I find it very related to this research, as it also refers to local features.
- I cannot understand why the authors refer to their approach as "mid-level features". I could not find anywhere in the paper an explanation on why the features after the transformer should be called like that. My understanding is that the LIT will leverage **all** the local features to produce the super-features. The super-features will incorporate then global context, and I wouldn't call a feature that incorporates information from all the image as mid-level... The authors should explain why this naming or remove it.

**Summary Of The Paper:**

The paper proposes a novel model, denoted as super-features, for image retrieval, with the following specific contributions: 1) features are extracted using an iterative attention module (the Local Feature Integracion (LIT) Transformer). Although similar modules have been proposed before in the literature [Locatello 2020], the authors reference them properly and their proposal have several modifications (decorrelation loss, learning the initial templates, residual connections). 2) The model is trained using contrastive losses on the local features, opposed to the more spread training procedure using constraints on the global embeddings. The paper presents solid experimental results in public datasets and compare them against reasonably chosen baselines.

**Summary Of The Review:**

Summing up, although the paper does not have a very strong novelty, its several contributions together with the solid experimental results in a relevant application domain are sufficient for acceptance. I found two minor issues that I think the authors should fix, but the rest is excellently written and presented, which is also valuable.

---

> ### Author Response · Authors · 2021-11-17
> **Author's Response**
>
> We want to thank the reviewer for a constructive review, positive comments and high score. We are happy they found
> our experimental results "solid", our ablation studies "complete" and our finding that the commonly used global/aggregated loss is not longer needed "interesting". We are also happy they found "several details on the architecture and the losses" to be novel and that our method is enjoying "significantly improved performance". Below we respond to their comments and update the manuscript accordingly.
>
> &nbsp;
>
> > _I miss Patch-NetVLAD [Hausler et al. CVPR21] in the list of references and the baselines.
> I am not expecting it to outperform the authors' approach, as Hausler et al. report worse performance than DELG, that is outperformed by this research._
>
> We thank the reviewer for pointing out this recent related work we missed. It is indeed relevant and we have added a reference to it in the related work section. Since the paper does not report results on any of the datasets we experiment on, it is hard to add it in the list of baselines we compare to. As the reviewer already notes, however, we would expect it to perform comparable to DELG, i.e., a method that the proposed FIRe outperforms.
>
> &nbsp;
>
> > _I cannot understand the authors refer to their approach as "mid-level features" [..] The super-features incorporate global context [..] The authors should explain why this naming._
>
> We thank the reviewer for their comment and for spotting that the rationale for our choice of terminology is missing. It is true that the proposed Super-features leverage context and potentially have access to information from the entire image. Yet, we empirically observe that each one of them learns to attend to multiple __localized__ areas of the image, and that many of them seem to be "excited" by "mid-level" patterns (see also Figure 1). Although each Super-feature is indeed a weighted sum of the features from all locations, in practice this sum is dominated by the features from only a small subset of locations. It is noteworthy that this is not the case for attention maps of  (self-)attention modules in general; this is prominent in our case due to the existence of $L_{attn}$ in Eq.(9) that explicitly minimizes the spatial correlation between attention maps (see also Figure 4). We thank the reviewer and we added a discussion on this in the updated manuscript.

---

> > ### Comment · Reviewer_hp4Y · 2021-11-18
> > **Thanks for the replies and clarifications**
> >
> > I would like to thank first the authors' replies and clarifications, to my review and also to the rest of the reviews.
> >
> > I see now more clearly the rationale for the "mid-level" features naming, thanks for the clear explanations. However, I will insist that I do not see it as entirely appropriate. If your model focuses on a set of multiple areas of an image, I would refer to it as a model of the relations between several local features, which might or might not be a higher-level feature. The authors also argue that this was only experimentally observed, which I do not see as a strong validation of the type of features extracted.

---

> > > ### Author Response · Authors · 2021-11-19
> > > **The 'mid-level terminology'**
> > >
> > > We thank the reviewer for a constructive discussion.
> > >
> > > We agree that the term 'mid-level features' is not perfectly suited to describe our features and consequently we updated the text again to make sure we only use it as scarcely as possible. We pondered it at length and this is still the best we could find to quickly convey the intuition behind what our features do (please see the details below). Because of this we kept it in the abstract (where we need to be concise and do not have enough space to give a more elaborate definition), and updated the rest of the text to not refer to our features as 'mid-level'. We hope this solution is satisfactory, and want to thank the reviewer again.
> > >
> > > Discussion on other possible terms that we considered:
> > >
> > > We'd like to communicate that i) Super-Features can aggregate low-level information into an intermediate (more semantic) representation and ii) we learn them in a way that minimizes their spatial redundancy, _i.e._ so that many of them are needed to describe an image. We think the term 'mid-level' conveys these intuitions, despite it's flaws.
> > >
> > > For other names, we considered the term 'regional features' but it implies even more that they are localized and tend to be used primarily for connected regions, and the term 'attentional features', that we found too vague and to not reflect the fact that we are spatially de-correlating the attention maps of those features. The term 'aggregated features' does not go well either with our explicit de-correlation, and even more problematic, may create a confusion with the ASMK matching phase.

---

> > > > ### Comment · Reviewer_hp4Y · 2021-11-23
> > > > **Thanks again for the discussion**
> > > >
> > > > Thanks again for the discussion, that helped once again to better understand your work.
> > > >
> > > > What you just did is what I would have done: if none of the terms is perfectly suitable it is better not using them much, although I agree with the authors that you still have to transmit the message. I agree that neither attentional nor aggregated nor regional fully fits your model, so the best option seems to be just conveying the intuition without making strong use of the term.
> > > >
> > > > Best regards,

---

### Official Review · Reviewer_1P44 · 2021-11-01

**Correctness:** 3
**Technical Novelty And Significance:** 2
**Empirical Novelty And Significance:** 3
**Recommendation:** 6
**Confidence:** 4

**Main Review:**

## Pros
1. The paper has a clear motivation. The concept of super features is very intuitive.
2. The majority of the design is reasonable. A transformer might be the best bet currently if we want to aggregate local features.
3. Some parts of the experiments are interesting. Fig 6 clearly show how FIRe differs from HOW in selecting local features from all scales. The major results in Tab. 3 are quite promising.

## Cons/Questions
1. In Eq. (4), the attention maps are normalized twice alongside the axes of both L and N. In conventional attention, normalizing the attention along the key entries (counting to L) would be enough. So why should we do it again along the query entries?
2. I think LIT implements a set transformer [a], at least a part of the set transformer, in the way of reducing and aggregating features according to a set of dummy queries in a QKV manner. I haven't seen any discussion on this. So maybe the authors can elaborate more on this and find more related articles. They should be very careful when claiming the novelty of LIT.
3. The proposed model follows HOW to measure image similarity using ASMK. Does it have to be ASMK? As the authors mentioned in Sec. 3.1, S is ordered (which is quite different from the case of HOW). Simply comparing two super features from two sets on the same entries is reasonable as they reflect the same latent topic defined by Q. Aggregating the similarities on all entries only requires a time complexity of $O(Ld)$. I believe ASMK undergoes a higher time complexity of $O(1000^2d)$ according to the implementation. Why should we go for a more complex solution besides 'a fair comparison'?

## Optional Questions
4. $L_{attn}$ seems to be the solver in this paper regarding the oversmoothness problem. This design is just too empirical. Do we have a better option here?
5. Do we have a better option to get multi-scale features than rendering the resized images multiple times? It seems this solution requires too much memory.

------
[a] Lee, Juho, et al. "Set transformer: A framework for attention-based permutation-invariant neural networks." International Conference on Machine Learning. PMLR, 2019.

**Summary Of The Paper:**

This paper introduces the concept of super features into image retrieval. The whole framework is named FIRe. The idea is to aggregate a set of features on the feature maps at different scales according to a set of query vectors representing different latent concepts. Hence, the proposed model somehow implements a transformer, which is called LIT in this paper.

To train this model, the authors define two losses. The first one is a triplet-like one executed on the super-feature granular. Some empirical criteria are established to select positive and negative super features. The second one decorrelates the super features from a single image.

ASMK becomes the measurement to compare two images when testing, which is a conventional option in comparing two groups of local features.

We see experiments on ROxford and RParis. The results are quite promising.

**Summary Of The Review:**

I conservatively give a borderline before the discussion phase. The motivation and intuition of this paper are quite nice, but some parts of the details are questionable. I think the overall quality of this paper is about ok, but it doesn't hit. Some parts of FIRe are not absolutely novel. The authors need to defend these designs to convince the readers.

------------------------------------------------------------------------

The authors' response satisfies me. Hence, I have changed my final rating.

---

> ### Author Response · Authors · 2021-11-17
> **Author's Response - Part 1**
>
> We thank the reviewer for a detailed and constructive review. We are happy they found our motivation "clear", the concept of the proposed super-features "very intuitive", parts of the experiments interesting, and our major results "quite promising". Below we respond to all their questions and comments and we have updated the manuscript accordingly. We hope the reviewer finds our responses reasonable and satisfying; we welcome further discussion.
>
>
> &nbsp;
>
> > _In Eq. (4), the attention maps are normalized twice alongside the axes of both L and N. In conventional attention, normalizing the attention along the key entries (counting to L) would be enough. So why should we do it again along the query entries?_
>
> Unlike in the conventional self-attention mechanism, the two inputs that contribute to our attention matrix $\alpha$ are not linear projections of the same input vector; normalization in both ways would therefore make sense as each dimension can be seen as the marginal probability of one over the other (e.g. how each location contributes to each super-feature and vice versa). It therefore makes sense to apply some form of bi-directional normalization function, i.e. a joint normalization function across rows and columns that would give a matrix of scores normalised over both dimensions. As there is no closed form function for this operation, one can only approximate it; Using a Softmax along one and L1 along the other dimension is a convenient approximation that is also widely used (e.g. in the Slot attention of Locatello et al. (2020)). Running the Sinkhorn-knopp algorithm, or iteratively and alternatively applying softmax at each dimension till convergent would be other ways of approximating this, but we chose to use softmax and L1, as it does not require an iterative process. We hope that this explanation clarifies this part for the reviewer. We agree that this was not clear from the text, so we updated the text and tried to clarify. Thank you!
>
> Following the reviewer's suggestion, we provide below results for a variant of our method without the L1 normalization; we observe a clear drop in performance.
>
> | Method            | SfM-120k | ROxford | ROxford | RParis | RParis |
> |-------------------|----------|---------|---------|--------|--------|
> |                   | val      | med     | hard    | med    | hard   |
> | FIRe w/o L1-norm  | 76.5     | 61.8    | 38.5    | 71.9   | 59.3   |
> | FIRe              | **89.7** | **81.8** | **61.2** | **85.3**   | **70.0**   |
>
>
> &nbsp;
>
> > _I think LIT implements a set transformer [a], at least a part of the set transformer, in the way of reducing and aggregating features according to a set of dummy queries in a QKV manner. I haven't seen any discussion on this. So maybe the authors can elaborate more on this and find more related articles. They should be very careful when claiming the novelty of LIT._
>
> We thank the reviewer for a relevant missing citation; we added it in the related work section.
> The reviewer makes a good point; indeed, same as the set transformer, we also consider the input as a set and apply no positional encoding; beyond that, however, the core architecture of our module is closer to the more recent DETR and slot attention architectures of Locatello et al. (2020) than the set transformer (we refer the reviewer to Section 5 where we explicitly discuss architectural differences to Locatello et al. (2020)). The design for our module was motivated by the fact that the slot attention mechanism did not scale to a large set of output features required _for the task of landmark retrieval_. Looking at the type and scale of tasks that the set transformer was evaluated on in Lee et al. (2019), we can assume that neither the set transformer architecture would work for our task out of the box.
>
> Let us also note that the way the proposed LIT module is used and especially _trained_ differs from all aforementioned approaches, including the set transformer: we are not only using a weakly-supervised loss on matching Super-features, but also add an extra loss directly diversifying the attention maps.

---

> > ### Author Response · Authors · 2021-11-17
> > **Author's Response - Part 2**
> >
> > > _The proposed model follows HOW to measure image similarity using ASMK. Does it have to be ASMK? [...] comparing two super features from two sets on the same entries is reasonable [...] ASMK undergoes a higher time complexity [...]Why should we go for a more complex solution besides 'a fair comparison'?_
> >
> > This is also a good point. We adopted ASMK as it was the top-performing matching strategy in recent related works and we simply switched from aggregating local to aggregating Super-features with ASMK. As the reviewer correctly notes, the order of Super-features is not leveraged by ASMK. During this discussion period, we tried having separate ASMK codebooks per Super-feature, but preliminary experiments did not show any gains out-of-the-box. Although we agree that it makes sense to utilize the power of Super-features also during matching, we believe that this is not a trivial task and better left to be explored in the future. We hope that the reviewer agrees that showing the strong performance gains over the state-of-the-art with a popular matching strategy is sufficient.
> >
> > Following the reviewer recommendation, we also evaluated Super-Features by directly computing the similarity between two images as the average of the similarities between their Super-features. Compared to ASMK, we observe an approximately 7\% performance drop when simply using the sum of dot product similarities between each corresponding Super-feature. We see many reasons for this: a) ASMK deals with occlusions better, i.e., with features that do match between two images showing the same landmark but with different parts of it; b) ASMK has a large (64K) vocabulary and is _selective_ and only matches features that are highly similar. On the other hand, forcing _every_ Super-feature pair to match, can be highly noisy; although corresponding, many such pairs should be discarded.
> > Note also that such a metric would require storing the $N$ Super-Features of dimension $d$ for each database image (single scale), which represents 32,768 floating numbers (131KB) per image for $N=256$ and $d=128$, i.e., not scalable.
> >
> > Beside the advantages mentioned above, ASMK is also an extremely memory and computationally efficient way of comparing sets of features.
> > Given a test image, each of the $N$ Super-Features are compared to and assigned to the $K$ ASMK cluster centroids-this process is irrespectively of the number of database images. On average, we obtain between 350 and 400 clusters with at least one assignment for each image with multi-scale testing.
> > For each assigned cluster, features assigned to it are aggregated and _binarized_. In terms of memory, what needs to be saved is thus 128 bits (16 bytes) per assigned cluster plus the index of this cluster (2 bytes for uint16 or 4 bytes  worse case).
> > Thanks to an inverted file structure, the similarity to all image databases can be efficiently estimated by looking only at the clusters with assignment for the query images (i.e., around 350-400 clusters among the 65,536 clusters we use).
> >
> > &nbsp;
> >
> > > $L_{attn}$ _seems to be the solver in this paper regarding the oversmoothness problem. This design is just too empirical. Do we have a better option here?_
> >
> > The loss we chose is indeed a common and straightforward way of diversifying the attention maps via minimizing the off-diagonal elements of the self-correlation matrix. We consider the fact that our method is able to reach state-of-the-art performance with such a straightforward and simple de-correlation loss is an _advantage_ of our method.
> >
> > &nbsp;
> >
> > > _Do we have a better option to get multi-scale features than rendering the resized images multiple times? It seems this solution requires too much memory._
> >
> > This is an interesting point, thank you. We did not consider something beyond what is in the common evaluation protocol for a fair comparison to related works; we believe any change in the protocol to be orthogonal and beyond the scope of our paper.
> >
> > We do think that reviewer asks an interesting question, though. In some sense, Super-features are inherently scale-invariant (see Figure 10) and we believe this to be the main reason behind the surprisingly large gains our method has over the global loss (Figure 9). Having access to the input in multiple resolutions, however, seems to still have a strong effect in terms of final performance. Given that a) the proposed method behaves differently from the previous methods with respect to scales, see Figure 6, b) single-scale performance gains over the global loss are huge, see Figure 9, and c) spatial averaging/subsampling can happen after the backbone, before the LIT module, one could envision a more efficient pipeline for multi-scale inputs. We thank the reviewer; although it is practically infeasible to run such experiments during the discussion period, we will consider adding any finding in the future. Let us note again that we consider any such improvement orthogonal to our main contribution.

---

> > > ### Comment · Reviewer_1P44 · 2021-11-27
> > > **Good response**
> > >
> > > The response from the authors makes sense. Though I still have a slight concern about the contribution overall, recalling the recent works in transformer and how they claim their novelty, I think this work does have its own merits. Hence I would raise my score to support the acceptance of this paper.

---

### Official Review · Reviewer_uoYN · 2021-11-03

**Correctness:** 4
**Technical Novelty And Significance:** 3
**Empirical Novelty And Significance:** 3
**Recommendation:** 8
**Confidence:** 5

**Main Review:**

=== Strengths ===

- LIT is an interesting new method for local feature extraction from images using a transformer-like architecture.
- While supervision is only needed on the image level, the loss is applied on the local feature level. In contrast, previous work on deep learning based local features applied the loss on aggregated local features.
- The ordered property of SuperFeatures helps interpret what each feature represents semantically.
- Rigorous ablation experiments clearly motivate the design and parameter choices.
- SuperFeatures consistently outperform the state of the art on the ROxford and RParis datasets.
- A good selection of relevant related work is cited.
- The paper is well-written, well-organized and easy to follow.

=== Weaknesses ===

- Some inconsistencies in the mathematical formulation of the method make it difficult to understand its details and would also make it difficult to reimplement the method. Details on this in the “Questions and suggestions on Sec. 3.1” section below.
- The computational cost of extracting SuperFeatures from an image is not discussed. This would be important for real-time image retrieval systems where features of the query image have to be extracted on-the-fly. It would be interesting to compare the speed of SuperFeature extraction to, e.g., DELG or HOW.

=== General questions and suggestions ===

- In the paper, image retrieval is only performed using ASMK. Have the authors tried other methods too? In particular, do SuperFeatures lend themselves to traditional spatial verification using RANSAC? If yes, it would be interesting to see results on that. If not, it would be helpful to clarify in the paper why this is not possible.
- Have the authors performed an ablation where the loss is applied on an aggregation of all local features? This could further motivate why applying the loss on the local feature level is beneficial.
- How sensitive is the performance of SuperFeatures to the initial set of templates? Are templates specific to a certain domain, e.g., landmarks or can they are transferred to, e.g., products or artwork?
- In Tab. 1, it would be interesting to also have the following rows: ”reciprocal+same”, “ratio only” and “same+ratio”.
- What is the value of N (number of SuperFeatures) in the experiments?
- The ablations show that matching only SuperFeatures with the same ID during training increases performance. However, in retrieval with ASMK, the ordered property of SuperFeatures is not taken advantage of. Have the authors tried an image retrieval method that matches only identical SuperFeature IDs?
- In Fig. 5, what are the numbers (1000, 2000) next to the curves?
- In the Related Work section, consider renaming the first subsection to “Image descriptors”
- In Related Work on image retrieval, consider citing GeM (Radenovic2018b), since it’s a popular method for feature aggregation that SuperFeatures are also compared against.
- In the second to last row of page 4, it seems a word is missing. Maybe it should be “the attention maps *tend* to be similar”

=== Questions and suggestions on Sec. 3.1 ===

- Are attention maps (\alpha_l) the rows or the columns of the matrix \alpha? Put differently, does an attention map tell me how well a single SuperFeature responds to each image location (like in Fig. 1), or do they tell me how much a single image location excites each SuperFeature?
- In the 1st equation in Eq. 4, should the final index be \alpha_N instead of \alpha_L? This seems to be inconsistent with the row above that says that there are N attention maps.
- It would help if the paper could specify the linear projection dimensions d_k, d_v and d_q.
- What are the dimensions of the three matrices in the second equation of Eq. 5? As far as I understand: \alpha is an (L x N) matrix as per Eq. 4, V(U) is an (N x L) matrix because it needs to have as many rows as \alpha has columns, Q is a (d x N) matrix since it has N templates of dimension d. However, if these are true, then the addition would not work since an (L x L) matrix would be added to a (d x N) matrix. If I understand correctly, the output of \psi(U, Q) needs to be a (d x N) matrix, meaning \alpha would need to have d rows and V(U) would need to have N columns. I’d appreciate it if the authors could point out where I’m wrong here.
- In the same equation, V(U) is a linear projection function applied to the set U. I think it is implied here that the elements of the set are concatenated into a matrix before applying the function. It would help if this were clarified in the text.
- The 2nd equation of Eq. 4 seems to divide vectors by vectors. I think there is a 2nd index missing in the divisor, i.e. it should be \alpha_{l,i}
Equations 4 and 5 skip the “t” superscript, as explained in footnote 2. I personally think that having the superscript in these equations would help with clarity.
- What is the motivation for applying softmax and L1 normalization to \alpha?
- In the 1st equation of Eq. 5, I’d suggest adding “Q^{t+1} = \phi(...)“
- I’d suggest using n instead of j as the running index for the templates, which would be consistent with using l as the running index for the local features.
- I’d suggest putting footnote 3 in the main text since it is an important detail.

**Summary Of The Paper:**

The paper proposes Local Feature Integration Transformer (LIT), a new architecture to extract mid-level local features from images that are used for image retrieval. The resulting features are called SuperFeatures and are shown to yield state-of-the-art performance on two standard image retrieval benchmarks. The paper further proposes a learning framework, called Feature Integration-based Retrieval (FIRe), to train LIT using a contrastive loss with only image-level supervision. In contrast to other deep learning based image features that apply the loss on the image embedding or aggregated local features, FIRe applies the loss on the local features directly. An interesting property of SuperFeatures is that they are ordered, meaning a SuperFeature with the same index i in two matching images will be localized on similar structures.

**Summary Of The Review:**

I am confident to suggest the paper for acceptance since it introduces a novel and interesting way of using transformers to train local image features that outperform the state of the art. The method is easy to train since only image-level labels are required, however the loss is applied in the local feature level, which is an interesting novelty.  Detailed ablations motivate the design of the method. My main concern are inconsistencies in the mathematical formulation of the method, which I am sure can be resolved for the camera-ready version.

---

> ### Author Response · Authors · 2021-11-17
> **Author's Response - Part 1**
>
> We thank the reviewer for their constructive feedback and detailed review. We are glad they found our method "interesting" and "new", that it "consistently outperformed the state of the art", and that our paper had "rigorous ablation experiments". We also warmly thank them for the detailed suggestions that led us to improve the text in Section 3.1. We answer to all questions below and have updated the text accordingly.
>
> &nbsp;
>
> > _computational cost of extracting SuperFeatures_
>
> We calculated the time required for extracting multi-scale features for 5000 images for HOW and FIRe. On our server, it took 157 seconds for HOW and 172 seconds for FIRe, _i.e._, an increase of 10\% on average. We added a discussion in the manuscript.
>
> &nbsp;
>
> >  _ablation where the loss is applied on an aggregation of all local features?_
>
> We report results when using a loss on the aggregation of all Super-features in Table 2 ($L_{global}$). This is similar to what HOW does for local features.
>
> &nbsp;
>
> > _Are templates specific to a certain domain, e.g., landmarks or can they be transferred to, e.g., products or artwork?_
>
> This is a great point, thank you. In our experiments, the templates are _learned_ together with the LIT module - we can therefore assume that they are adapted to the task at hand. To explore the sensitivity to the initial templates, we run a variant of FIRe where the initializations are not finetuned, but instead frozen to the values after pretraining on ImageNet. We report results in the table below as well as in Appendix A.5 of the updated manuscript. The performance achieved by that FIRe model is on the par with the one of learned templates. We believe that this ablation shows that the initial templates are up to some point transferable to other tasks. We understand that ImageNet to landmarks is not a large domain shift, but we believe it is beyond the scope of an authors' response to experiment on new datasets and different domains like products or artworks; we do think that this is however an interesting exploration for the future.
>
> | Template Initialization          | SfM-120k | ROxford | ROxford | RParis | RParis |
> |----------------------------------|----------|---------|---------|--------|--------|
> |                                  | val      | med     | hard    | med    | hard   |
> | Frozen from ImageNet pretraining | 89.5     | 81.3    | 59.6    | **85.3**   | **70.2**   |
> | Finetuned for landmark retrieval | **89.7**     | **81.8**    | **61.2**    | **85.3**   | 70.0   |
>
> &nbsp;
>
> > _[In Tab. 1] have the following rows: ”reciprocal+same”, “ratio only” and “same+ratio”._
>
> We added an extended table in the Appendix.
>
> &nbsp;
>
> > _What is the value of N in the experiments?_
>
> We use N=256 for all experiments (see also Sec. 4.1). We study the impact of this hyper-parameter in Table 7a of the Appendix.
>
> &nbsp;
>
> > _[In ASMK], the ordered property of SuperFeatures is not taken advantage of. Have the authors tried a retrieval method that matches only identical SuperFeature IDs?  [...] do SuperFeatures lend themselves to spatial verification using RANSAC?_
>
> This is a good point. We adopted ASMK as it was the top-performing matching strategy in recent related works and we simply switched from aggregating local to aggregating Super-features with ASMK. As the reviewer correctly notes, the order of Super-features is not taken advantage of by ASMK. During this discussion period, we tried having separate ASMK codebooks per Super-feature, but preliminary experiments did not show any gains out-of-the-box. Although we agree that it makes sense to utilize the power of Super-features also during matching, we believe that this is not a trivial task and better left to be explored in the future. We hope that the reviewer agrees that showing the superiority of our features with a popular and state-of-the-art matching strategy is sufficient.
>
> Running RANSAC on Super-features is far from trivial. As illustrated in Figure 1, they are not really easy to localize and estimate a transformation from. Although there are things one could try in that direction (e.g. binarization or per-Super-feature RANSAC) we believe these to be out-of-scope for this paper.
>
> &nbsp;
>
> > _In Fig. 5, what are the numbers (1000, 2000)?_
>
> Solid markers emphasize the most commonly used settings for the number of features aggregated with ASMK (1000 and 2000). Curves are formed by varying the number of features that are extracted per image; this set is then aggregated with ASMK. We experiment with {200, . . . , 5000} extracted features. Once extracted, each feature is associated to an ASMK centroid using nearest neighbors, and features associated to the same centroid are aggregated together. In the end, only a subset of the ASMK centroids have associated features and this is what is needed to perform matching with ASMK. We have improved the caption in the updated manuscript.

---

> > ### Author Response · Authors · 2021-11-17
> > **Author's Response - Part 2**
> >
> > We want to sincerely thank the reviewer for their detailed comments on Section 3.1; we think that their comments helped us significantly improve the clarity of that section. We answer all questions below, and updated the manuscript accordingly.
> >
> > &nbsp;
> >
> > > _What is the motivation for applying softmax and L1 normalization to $\alpha$?_
> >
> > Unlike in the self-attention mechanism, the two inputs that contribute to our attention matrix $\alpha$ are not linear projections of the same input vector; normalization in both ways would therefore make sense as each dimension can be seen as the marginal probability of one over the other (e.g. how each location contributes to each Super-feature and vice versa). It therefore makes sense to apply some form of bi-directional normalization function, i.e. a joint normalization function across rows and columns that would give a matrix of scores normalized over both dimensions (we also refer the reviewer to our response to Reviewer 1P44 for an ablation when using only one-way normalization). As there is no closed form function for this operation, one can only approximate it; Using a Softmax along one and L1 along the other dimension is a convenient approximation that is also widely used (e.g. in the Slot attention of Locatello et al. (2020)). Running the Sinkhorn-knopp algorithm, or iteratively applying softmax at each dimension till convergence would be other ways of approximating this, but we chose to use softmax and L1, as it does not require an iterative process. We hope that this explanation clarifies this part; we also refer to the reviewer to the related part in the response to Reviewer 1P44 for results without the L1 normalization. We further updated and clarified the text. Thank you!
> >
> > &nbsp;
> >
> > > _Are attention maps ($\alpha_l$) the rows or the columns of the matrix $\alpha$? [...] In the 1st equation in Eq. 4, should the final index be $\alpha_N$ instead of $\alpha_L$?_
> >
> > We are sorry for the confusion; it is our fault, Eq(4) was not written in the most intuitive way, while we overloaded the notation $\alpha_i$ which was referring to both the rows and the columns of $\alpha$ at different times (Eqs.(4) and (9), respectively).
> >
> > Attention maps are the columns of the matrix $\alpha$. In the updated manuscript we have a) updated the left-most equation to be a column vector, and b) clearly state in the equation premise that the attention maps are the __columns__ of matrix $\alpha$ (and not the row vectors $\alpha_l$) c) altered the notation we use when referring to the attention maps in Eq.(9). Since it is easier to mathematically present this bi-directional normalization per location $L$ we left the rest of the equations as is. We hope that the reviewer now finds the updated Eq(4) clear. Any comments or further discussion is welcome.
> >
> > &nbsp;
> >
> > > _It would help if the paper could specify the linear projection dimensions $d_k$, $d_v$ and $d_q$._
> >
> > We mention that $d_k=d_v=d_q=1024$ in the "implementation details" paragraph in Section 4, but we agree that it would be clearer to also mention this in Section 3.1; we updated the text accordingly.
> >
> > &nbsp;
> >
> > > _In the 1st equation of Eq. 5, I’d suggest adding “Q^{t+1} = \phi(...)“ _
> > > _ I’d suggest using n instead of j as the running index for the templates_
> >
> > Thank you for great suggestions - we implemented both in the updated manuscript.
> >
> > &nbsp;
> >
> > > _What are the dimensions of the three matrices in the second equation of Eq. 5?_
> >
> > $\alpha$ is $L \times N$, $V(U)$ is $d_v \times L$, the product of the latter with $\alpha$ is along the local feature dimension and the result is of size $N \times d_v$.
> >
> > &nbsp;
> >
> > > _In the same equation, V(U) is a linear projection function applied to the set U. I think it is implied here that the elements of the set are concatenated into a matrix before applying the function._
> >
> > This is true in terms of implementation; in mathematical terms, each local feature is treated independently (there is no mixing between local features, they are simply processed with parallel linear projections), so it can still be viewed as a function over a set.
> >
> > &nbsp;
> >
> > > _The 2nd equation of Eq. 4 seems to divide vectors by vectors._
> >
> > This division is to be understood as a broadcasted division between two vectors of equal dimensions.
> > The sum $\sum_{i=1}^L \hat{\alpha}_i$ contains vectors of identical dimensions, so a vector itself. The sum is over the local-feature index, so each element of the result gives, for a fixed Super-feature index, the sum of scores over all local features. This sum is used to normalize $\hat{\alpha}_l$. Thus when summing all $\alpha_l$ vectors, the result is a vector full of ones: each super-feature pays an equal amount of attention to the set of local features.

---

> > > ### Comment · Reviewer_uoYN · 2021-11-22
> > > **Thank you for the clarifications and improvements to the paper!**
> > >
> > > I am glad I could help clarify this section and would like to thank the authors for their detailed explanations and their improvements to the manuscript. I re-read the section and find it to be much easier to follow now. See below for my responses:
> > >
> > > * **L1 and softmax**: Thank you for the detailed explanation and paper reference. This clarified the motivation for me.
> > > * **Attention map notation**: I think the updates to the text make it much clearer, however I still briefly got confused: Above equation 4, the paper mentions that attention maps are the columns of $\alpha$. But Eq. 4 then introduces $\alpha$ as a column vector of row vectors $\alpha_1$ to $\alpha_L$. I think this may initially lead readers to think that these row vectors are the attention maps, which immediately conflicts with the statement above the equation. Only later in Eq. 9, the paper introduces the notation $\tilde{\alpha}_1$ to $\tilde{\alpha}_N$ for the attention maps, which clarifies this confusion. A suggestion to alleviate this would be to introduce a name for the rows of $\alpha$ as well, say, "response maps". Then, before Eq. 4, the paper could introduce the dual interpretation of $\alpha$ as a column vector of response maps $\alpha_l$ or a row vector of feature maps $\tilde{\alpha}_n$.
> > > * **Matrix dimensions in Eq. 5**: Thanks for the clarification. I'd suggest swapping the places of $\alpha$ and $V(\mathcal{U})$ so the dimensions match for the multiplication: $\psi(\mathcal{U}; \mathcal{Q}) = V(\mathcal{U}) \cdot \alpha + \mathcal{Q}$.
> > > * **V(U) as a set function**: I agree that V could be viewed as a set function. This also causes the matrix multiplication to be overloaded, but I think the current notation is easier on the reader than formally concatenating the set elements into a matrix.
> > > * **Division by vector in Eq. 4**: Thanks for the explanation! I think this is slightly nonstandard notation, but together with the explanation in the text below, it should be clear to the reader.

---

> > > > ### Author Response · Authors · 2021-11-23
> > > > **Further updates**
> > > >
> > > > We agree on all points and will incorporate the reviewer's suggestions in an updated version. We sincerely thank the reviewer once again for high-quality constructive feedback that made the presentation of our method clearer

---

> > ### Comment · Reviewer_uoYN · 2021-11-22
> > **Thank you for the detailed response!**
> >
> > I’d like to thank the authors for taking care to address my questions and concerns with great attention and detail!
> >
> > * **ImageNet Ablation**: The ablation indeed suggests that they might be transferable to other tasks. Thank you for this additional experiment.
> > * **Using other matching techniques**: I think there could be some potential in finding a matching technique that makes better use of the properties of SuperFeatures, but I appreciate that this is not trivial and agree that this should be left as future work. Thank you for your attempts at modifying ASMK in this regard.
> > * **Fig. 5**: Thanks for the clarification and for updating the caption.

---

### Decision · Program_Chairs · 2022-01-20

**Decision:**

Accept (Poster)

**Comment:**

The paper proposes a model for large-scale image retrieval. Unlike previous work that rely on local features, the proposed method aggregates local features into the so-called Super-features to improve their discriminability and expressiveness. To do so, the method proposes an iterative attention module (Local Feature Integracion Transformer, LIT), that outputs an ordered set of such features. By exploiting the fact that features are ordered, the paper proposes a contrastive loss on Super-features that match across images. The paper presents a thorough empirical evaluation on several publicly available datasets including relevant baselines.

Overall the paper is well written and the empirical results are strong (including detailed ablations that motivate the design of the method). All reviewers and the AC appreciate the idea of applying the contrastive training at local feature level while only requiring image-level labels.

Reviewer hp4Y points out that the proposed LIT is not particularly novel, but previous work are properly cited. Also this is not a major issue given that the motivation is very clear, it is well executed and the empirical results are strong.

Reviewer uoYN had initial concerns regarding inconsistencies in the mathematical formulation of the method, which were resolved in a detail (and constructive) discussion with the authors.

All reviewers recommend accepting the paper, three of which consider the contribution to be strong. The AC agrees with this assessment and recommends accepting the paper.